# A two-step mechanism for the inactivation of microtubule organizing center function at the centrosome

Jérémy Magescas, Jenny C Zonka, Jessica L Feldman*

Department of Biology, Stanford University, Stanford, United States

**Abstract** The centrosome acts as a microtubule organizing center (MTOC), orchestrating microtubules into the mitotic spindle through its pericentriolar material (PCM). This activity is biphasic, cycling through assembly and disassembly during the cell cycle. Although hyperactive centrosomal MTOC activity is a hallmark of some cancers, little is known about how the centrosome is inactivated as an MTOC. Analysis of endogenous PCM proteins in *C. elegans* revealed that the PCM is composed of partially overlapping territories organized into an inner and outer sphere that are removed from the centrosome at different rates and using different behaviors. We found that phosphatases oppose the addition of PCM by mitotic kinases, ultimately catalyzing the dissolution of inner sphere PCM proteins at the end of mitosis. The nature of the PCM appears to change such that the remaining aging PCM outer sphere is mechanically ruptured by cortical pulling forces, ultimately inactivating MTOC function at the centrosome.
DOI: https://doi.org/10.7554/eLife.47867.001

*For correspondence:
feldmanj@stanford.edu

Competing interests: The authors declare that no competing interests exist.

## Introduction

Numerous cell functions such as transport, migration, and division are achieved through the specific spatial organization of microtubules imparted by microtubule organizing centers (MTOCs). The best-studied MTOC is the centrosome, a membrane-less organelle composed of two barrel-shaped microtubule-based centrioles surrounded by a cloud of pericentriolar material (PCM). Microtubules at the centrosome are mainly nucleated and localized by complexes within the PCM, which generate a radial array of microtubules in dividing animal cells and some specialized cell types such as fibroblasts.

The PCM is a central hub for the regulation of a number of cellular processes including centriole duplication, ciliogenesis, cell cycle regulation, cell fate determination, and microtubule organization (*Chichinadze et al., 2013*; *Fry et al., 2017*; *Stubenvoll et al., 2016*). In *Drosophila* and human cell lines, PCM proteins including a subset of scaffolding proteins are organized in cumulative layers ultimately recruiting microtubule nucleation and organization factors, such as the conserved microtubule nucleating γ-tubulin ring complex (γ-TuRC) (*Fu and Glover, 2012*; *Lawo et al., 2012*; *Mennella et al., 2012*). In *C. elegans,* the PCM is much simpler in composition, built from the interdependent recruitment of two scaffolding proteins, SPD-2/CEP192 and SPD-5, the functional homologue of CDK5RAP2/Cnn (*Hamill et al., 2002*; *Kemp et al., 2004*). Together with the highly conserved kinase AIR-1/Aurora-A, SPD-2 and SPD-5 are required to localize γ-TuRC, which in *C. elegans* is composed of TBG-1/γ-tubulin, GIP-1/GCP3, GIP-2/GCP2 and MZT-1/MZT1 (*Bobinnec et al., 2000*; *Hamill et al., 2002*; *Hannak et al., 2002*; *Kemp et al., 2004*; *Lin et al., 2015*; *Oakley et al., 2015*; *Sallee et al., 2018*). γ-TuRC and AIR-1 have been shown to together be required to build microtubules at the centrosome in the *C. elegans* zygote (*Motegi et al., 2006*). Additionally, the major role of the γ-TuRC at the PCM in *C. elegans* might be to anchor microtubules at the periphery as loss of γ-TuRC results in microtubules distributed throughout the PCM (*O'Toole et al., 2012*).

**eLife digest** New cells are created when existing cells divide, a process that is critical for life. A structure called the spindle is an important part of cell division, helping to orient the division and separate parts of the old cell into the newly generated ones. The spindle is built using filamentous protein structures called microtubules which are arranged by microtubule organizing centers (or MTOCs for short). In animals, an MTOC forms at each end of the spindle around two structures called centrosomes.

A network of proteins called the pericentriolar material (PCM) form around centrosomes, converting them into MTOCs. The PCM grows around centrosomes as a cell prepares to divide and is removed again afterward. Enzymes called kinases are important in controlling cell division and PCM assembly; they are opposed by other enzymes known as phosphatases. The processes involved in organization and removal of the PCM are not well understood.

The microscopic worm *Caenorhabditis elegans* provides an opportunity to study details of cell division in a living animal. Magescas et al. used fluorescent labels to view proteins from the PCM under a microscope. The images showed two partially overlapping spherical parts to the PCM – inner and outer. Further examination revealed that the inner PCM is maintained by a careful balance of kinase and phosphatase activity. When kinases shut down at the end of cell division, the phosphatases break down the inner PCM. By contrast, the outer PCM is physically torn apart by forces acting through the attached microtubules.

Future work will seek to examine which proteins are specifically affected by phosphatases to identify the key regulators of PCM persistence in the cell and to reveal the proteins needed for MTOC activity at the centrosome. Since poor MTOC regulation can play a part in the growth and spread of cancer, this could lead to targets for new treatments.

DOI: https://doi.org/10.7554/eLife.47867.002

In addition to the γ-TuRC, several other microtubule regulating proteins are recruited to the PCM to promote its microtubule organizing center function through the stabilization and growth of microtubules. The conserved complex of ZYG-9/XMAP-215/Alp14, a processive microtubule polymerase (*Matthews et al., 1998*; *Thawani et al., 2018*), and TAC-1/TACC/Alp7 promotes microtubule polymerization (*Bellanger and Gönczy, 2003*; *Bellanger et al., 2007*). This complex is also involved in microtubule nucleation as has been recently shown in yeast and Xenopus egg extract (*Flor-Parra et al., 2018*; *Thawani et al., 2018*). In addition. the microtubule stabilizing and nucleation-promoting factor TPXL-1/TPX2 also localizes to the PCM where it interacts with and activates AIR-1 (*Bayliss et al., 2003*; *Zhang et al., 2017*). Although the pathways required to build the PCM are largely known in *C. elegans*, the organization of proteins within the PCM has been generally unexplored. One notable exception is that microtubules have been shown to localize to the periphery of the PCM by electron microscopy (*O'Toole et al., 2012*). As TBG-1 is found throughout the PCM, these studies suggest the existence of different pools of TBG-1 at the PCM, with an active population at the periphery that organizes microtubules.

The centrosome is not a static organelle; during each cell cycle, MTOC activity at the centrosome is massively increased to ultimately build the mitotic spindle (*Dictenberg et al., 1998*; *Woodruff et al., 2014*). This increase in centrosomal MTOC activity relies on the recruitment of PCM proteins to the centrosome, a process that is controlled by the concentration and availability of PCM proteins and their phosphorylation by mitotic kinases (*Decker et al., 2011*; *Wueseke et al., 2014*; *Wueseke et al., 2016*; *Yang and Feldman, 2015*). Three main kinases are involved in the regulation of PCM activity: CDK-1/CDK1, PLK-1/PLK1 and AIR-1/Aurora A (*Pintard and Archambault, 2018*). CDK-1 acts as the main driver of PLK-1 and AIR-1 activity, which likely directly phosphorylate PCM proteins to promote PCM assembly (*Woodruff et al., 2014*). During mitotic exit, MTOC activity of the centrosome rapidly decreases, marked by the reduction of the PCM and microtubule association. This cycle of centrosomal MTOC activity continues every cell cycle, but can also be naturally discontinued during cell differentiation when MTOC function is often reassigned to non-centrosomal sites (*Sanchez and Feldman, 2017*). Although the mechanisms controlling PCM disassembly have been relatively unexplored, inhibition of CDK activity can drive precocious PCM disassembly and

inhibition of the PP2A phosphatase LET-92 perturbs SPD-5 removal from the centrosome, suggesting that phosphatase activity could be more generally required for the inactivation of MTOC function at the centrosome (*Enos et al., 2018*; *Yang and Feldman, 2015*). Additionally, stabilization of CDK1 activity has been shown to inhibit PCM disassembly and promote PCM maintenance (*Rusan and Wadsworth, 2005*). Although kinase and phosphatase activity are implicated in this MTOC cycle, an understanding of how and when these factors act to inactivate MTOC function at the centrosome and whether all PCM proteins behave in the same manner during disassembly in vivo is currently lacking.

The inactivation of MTOC activity of the centrosome is likely critical in a number of cellular and developmental contexts. For example, asymmetric cell division is often associated with unequal PCM association at the mother vs. daughter centrosome and terminal differentiation of murine cardiomyocytes and keratinocytes has been linked to centrosome inactivation (*Cheng et al., 2011*; *Conduit and Raff, 2010*; *Muroyama et al., 2016*; *Zebrowski et al., 2015*). In an extreme example, female gametes in a range of organisms completely eliminate centrosomes and this elimination can be a critical step in gametogenesis (*Borrego-Pinto et al., 2016*; *Lu and Roy, 2014*; *Luksza et al., 2013*; *Mikeladze-Dvali et al., 2012*; *Pimenta-Marques et al., 2016*). Moreover, hyperactive MTOC function at the centrosome has been linked to several types of epithelial cancers and invasive cell behavior, and is a hallmark of tumors (*Godinho and Pellman, 2014*; *Lingle et al., 1998*; *Pihan, 2013*; *Pihan et al., 2001*; *Salisbury et al., 1999*). Despite the clear importance of properly regulating MTOC activity, little is known about the mechanisms that inactivate MTOC function at the centrosome, either what initiates the removal of PCM and microtubules during the cell cycle or what keeps them off the centrosome in differentiated cells.

To better understand how MTOC activity is regulated at the centrosome, here we investigate the localization and dynamics of endogenously tagged PCM proteins in the *C. elegans* embryo. We find that *C. elegans* PCM is composed of overlapping spheres of proteins similar to what has been observed in other systems, with SPD-5 and γ-TuRC occupying distinct regions from known binding partners SPD-2 and MZT-1, respectively. Live imaging of PCM components at the end of mitosis revealed two phases of disassembly, beginning with the gradual dissolution of PCM proteins such as PLK-1, SPD-2, TAC-1, and MZT-1, followed by the rupture of the remaining PCM proteins ZYG-9, SPD-5, γ-TuRC, TPXL-1, and AIR-1 into microtubule associated packets. Using pharmacological and genetic perturbations, we found a role for phosphatases in PCM disassembly throughout mitosis, opposing CDK activity during PCM assembly and catalyzing PCM dissolution once CDK activity naturally dissipated. Cell fusion and RNAi experiments indicated that the nature of the remaining PCM was transformed and mechanically cleared from the centrosome by cortical pulling forces. Delay in PCM removal impacted subsequent centriole separation and PCM maturation in the next cell cycle. These data indicate that the inactivation of MTOC function at the centrosome involves a regulated two-step process of PCM disassembly, the timing of which is critical to the developing embryo.

## Results

### *C. elegans* PCM is organized into an inner and outer sphere

In order to better understand how PCM proteins behave during disassembly, we first characterized the spatial organization of the PCM during mitosis in the ABp cell of the 4-cell *C. elegans* embryo. ABp has relatively large centrosomes oriented during mitosis along the left-right axis of the embryo, with one of the centrosomes positioned very close to the coverslip in an end-on orientation (*Figure 1A*). We analyzed the localization of endogenously-tagged PCM proteins immediately after nuclear envelope breakdown (NEBD) in the ABp cell (*Figure 1A*). At this time, the centrosome still functions as an MTOC, actively growing and organizing microtubules (*Figure 1A*).

We assessed the localization of the centriole component SAS-4, the PCM scaffolding proteins SPD-2 and SPD-5, the γ-TuRC components GIP-1 and MZT-1, the mitotic kinases AIR-1 and PLK-1, and the microtubule associated proteins ZYG-9, TAC-1 and TPXL-1 (*Figure 1B*, *Figure 1—figure supplement 1*). As expected, the centrioles sit at the center of the centrosome (*Figure 1B–C*, *Figure 1—figure supplement 1B and D*) surrounded by PCM proteins which formed ordered layers of protein localization. SPD-2 and SPD-5 localization at the PCM is co-dependent (*Hamill et al., 2002*; *Kemp et al., 2004*; *Pelletier et al., 2004*), however these proteins displayed distinct outer

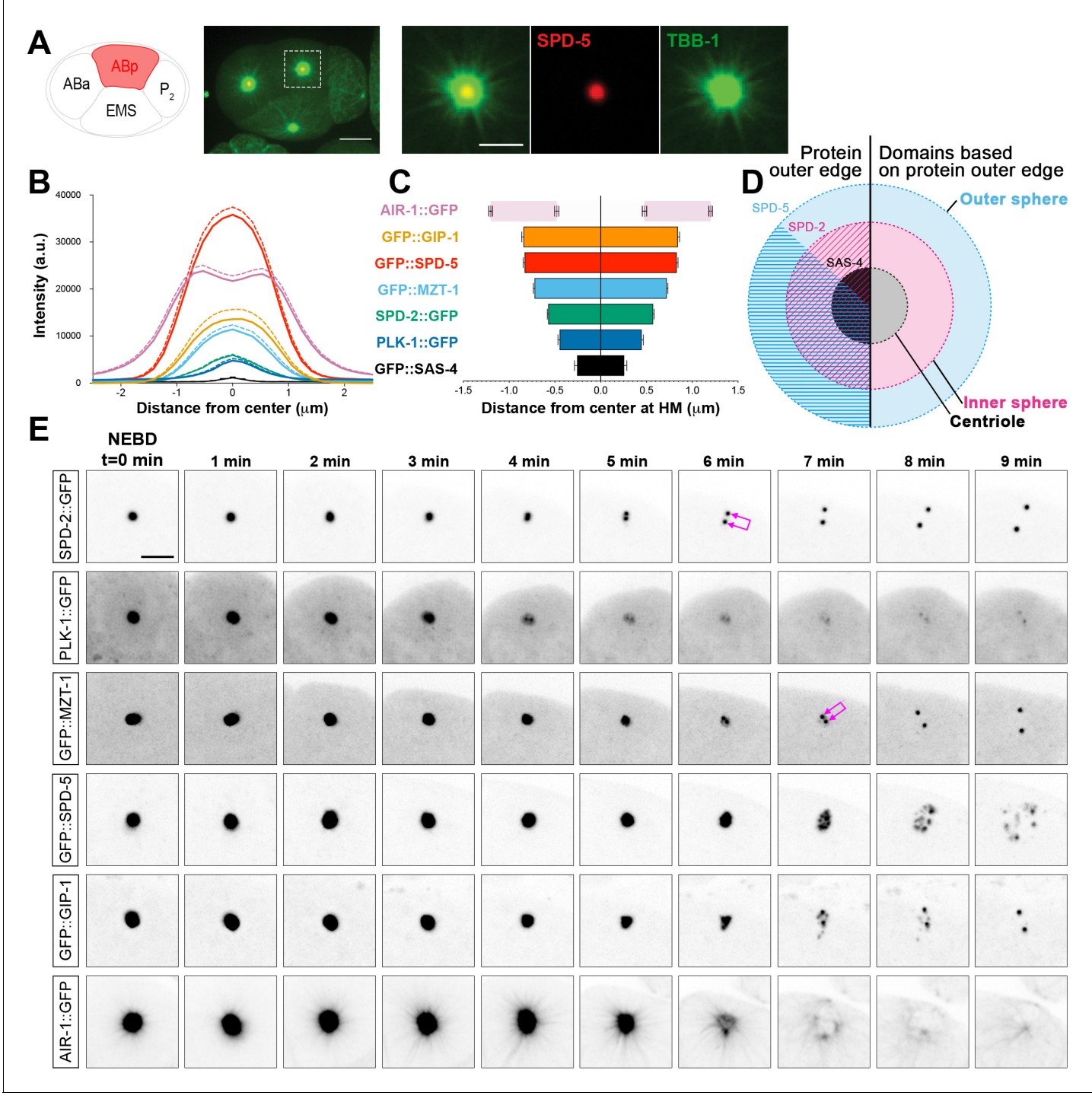

**Figure 1.** *C. elegans* PCM is organized into layered spheres that disassemble using different behaviors, see also *Figure 1—figure supplement 1*, *Figure 1—figure supplement 2*, *Videos 1–4*. (**A**) Left: Cartoon representing the *C. elegans* 4-cell stage embryo with ABp in red. Right: 7.5 μm z-projection from a live *pie-1p*::GFP::TBB-1/β-tubulin (green); tagRFP::SPD-5 (red) expressing embryo showing cell division in ABa and ABp. Note that these cells have a synchronized cell division and start dividing earlier than EMS or P2. Insets: Enlargement of ABp centrosome showing microtubules (green) organized around the centrosome (SPD-5, red). Scale bar, 5 μm. (**B**) Average pixel intensity profile across the ABp centrosome at NEBD: GFP:: GIP-1 (orange, n = 18), GFP::SPD-5 (red, n = 18), AIR-1::GFP (magenta, n = 19), GFP::MZT-1 (light blue, n = 21), SPD-2::GFP (green, n = 21), PLK-1::GFP (blue, n = 15), GFP::SAS-4 (black, n = 19). Bold line represents the mean, dotted lines represent standard error of the mean (s.e.m.). (**C**) Average distance from center at half maximum (HM) pixel intensity for each protein in B: SAS-4: −0.25–0.25 ± 0.06 μm, n = 19; PLK-1: −0.45–0.45 ± 0.02 μm, n = 15; SPD-2: −0.57–0.57 ± 0.01 μm, n = 21; MZT-1: −0.72–0.72 ± 0.01 μm, n = 21; SPD-5: −0.83–0.83 ± 0.01 μm, n = 18; GIP-1: −0.84–0.84 ± 0.02 μm, n = 18; AIR-1 inner bars are the internal edge of the toroid: −0.48–0.48 ± 0.02 μm, n = 19; AIR-1 outer bars are the external edge of the toroid: −1.20–

*Figure 1 continued on next page*

*Figure 1 continued*

1.20 ± 0.02 μm, n = 19. (D) Cartoon representing the organization of the centrosome based on the boundary of SAS-4 (black, 'centriole'), SPD-2 (cyan, 'inner sphere'), and SPD-5 (magenta, 'outer sphere'. (E) Time-lapse analysis of the disassembly of each protein analyzed in B, C and D starting at NEBD (t = 0 min) and imaged every minute for 9 min. Image LUTs have been scaled to their respective 7 min timepoint in order to demonstrate the packets observed during disassembly. Note that in some images, the two centrioles and the corresponding newly forming centrosomes become apparent (joined magenta double arrows) following removal of that protein from the PCM. Scale bar, 10 μm.

DOI: https://doi.org/10.7554/eLife.47867.003

The following figure supplements are available for figure 1:

**Figure supplement 1.** Methods for quantifying PCM width.

DOI: https://doi.org/10.7554/eLife.47867.004

**Figure supplement 2.** Time-lapse analysis of TAC-1, ZYG-9 and TPXL-1 during disassembly compared to SPD-2 and SPD-5.

DOI: https://doi.org/10.7554/eLife.47867.005

localization boundaries within the PCM; both SPD-2 and SPD-5 localized to a more proximal region surrounding the centrioles (distance from center at half maximum intensity for SPD-2: −0.575–0.575 ± 0.02 μm; 77.8 ± 0.8% of total SPD-5 overlapping with SPD-2 in this region), and SPD-5 extended to a more distal region lacking SPD-2 (distance from center at half maximum intensity for SPD-5: −0.83–0.83 ± 0.03 μm; *Figure 1B–C*). Based on the outer edge of these two matrix proteins, we divide the PCM into an 'inner' and 'outer' sphere, with the smaller inner sphere defined by the outer edge of SPD-2 localization and the larger outer sphere defined by the outer edge of SPD-5 localization (*Figure 1D*). PLK-1 showed the most restricted localization, occupying a relatively proximal localization in the inner sphere (*Figure 1B,C,E*). GIP-1 localization was indistinguishable from SPD-5, extending into the outer sphere (*Figure 1B–C*, *Figure 1—figure supplement 1B and D*), however another γ-TuRC component MZT-1 showed an intermediary localization, extending only partially into the outer sphere (*Figure 1B–C* and *Figure 1—figure supplement 1B,D*). TAC-1 and ZYG-9 also shared this intermediate localization pattern (*Figure 1C*, *Figure 1—figure supplement 1B–D*). Finally, the localization of AIR-1 was mainly restricted to the outer sphere, forming a toroid as previously reported (*Hannak et al., 2001*) and a complimentary localization pattern to PLK-1. As expected, TPXL-1 and AIR-1 co-localized, consistent with the fact that TPXL-1 is important for AIR-1 recruitment to the centrosome and microtubules (*Toya et al., 2011*). Both TPXL-1 and AIR-1 localization extended further than the boundary of SPD-5 and GIP-1 (*Figure 1—figure supplement 1B–D*).

Based on these observations, we conclude that the PCM has a layered structure with an inner sphere delimited by SPD-2 localization (*Figure 1D*) that also localizes SPD-5, PLK-1, ZYG-9, TAC-1, γ-TuRC components, AIR-1, and TPXL-1, and an outer sphere delimited by SPD-5 localization that also contains ZYG-9, TAC-1, GIP-1, MZT-1, AIR-'1, and TPXL-1 (*Figure 1D*). Although our imaging approach did not allow us to resolve toroidal localization patterns of the majority of the PCM proteins we analyzed, the boundaries of PCM protein localization follows the general pattern of the predicted orthologs in *Drosophila* and human cells (*Fu and Glover, 2012*; *Lawo et al., 2012*; *Mennella et al., 2012*; *Mennella et al., 2014*). This localization pattern is also noteworthy as SPD-5 and GIP-1 are found in a region lacking known binding partners SPD-2 and MZT-1, respectively.

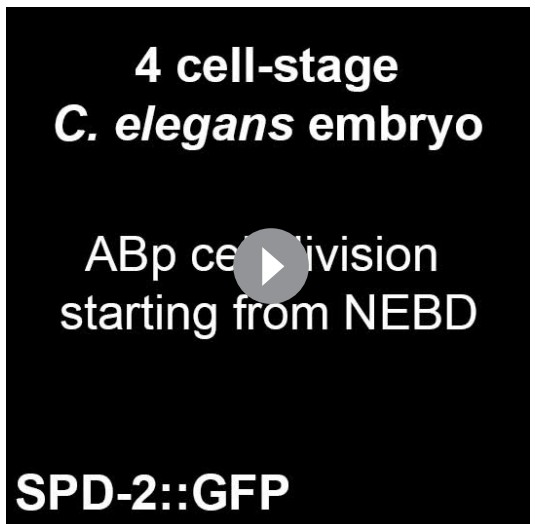

**Video 1.** Centrosome disassembly in the ABp cell in a 4-cell embryo expressing endogenous SPD-2::GFP. Scale bar, 5 μm.

DOI: https://doi.org/10.7554/eLife.47867.007

## PCM proteins disassemble with different behaviors

Based on their distinct localization within the PCM, we hypothesized that different PCM

proteins would disassemble with different kinetics and behaviors. To test this hypothesis, we examined the dynamics of disassembly of each of the endogenously-tagged PCM proteins described above by live-imaging in the ABp cell beginning at NEBD (*Figure 1E* and *Figure 1—figure supplement 2A*). SPD-2 (*Video 1*), MZT-1 (*Video 2*), TAC-1, and PLK-1 displayed similar disassembly behaviors, leaving the centrosome by gradual 'dissolution' over time and eventually only remaining at the two centrioles that will duplicate and mature into new centrosomes (*Figure 1E* and *Figure 1—figure supplement 2A*). In contrast, SPD-5 (*Video 3*), GIP-1 (*Video 4*), ZYG-9, AIR-1, and TPXL-1 initially showed some gradual disassembly, however the structure containing these proteins then appeared to 'rupture' and fragment into 'packets' that were distinct from the centrioles (*Figure 2A–C*). These sub-PCM packets localized SPD-5, GIP-1 (*Figure 2A*, *early packets*), microtubules (*Figure 2B*, *early packets*), AIR-1 (*Figure 2C*, *early packets*), and TPXL-1, but neither SPD-2 nor MZT-1 (*Figure 2E*, see below). Intriguingly, packets appeared to retain MTOC potential as EBP-2/EB1 comets, a marker of growing microtubule plus ends, dynamically moved from the SPD-5/GIP-1 foci (*Figure 2D*). The packets appeared to be further disassembled in the cytoplasm following their removal from the PCM, with GIP-1 and microtubules first losing their association, followed by SPD-5 (*Figure 2A–C*, *late packet*s, *Figure 2F*).

To gain a better sense of the timing of the disassembly of the different PCM proteins, we imaged each protein in combination with SPD-5. SPD-2 (*Figure 3A*) and MZT-1 (*Figure 3B*) showed a gradual decrease in intensity, beginning at 2 (2.20 ± 0.13 min, n = 10) or 3 min (3.00 ± 0.27 min, n = 8) post-NEBD, respectively, and a decrease in PCM volume beginning 3 min post-NEBD (SPD-2: 3.00 ± 0.21 min, n = 10; MZT-1: 2.88 ± 0.23 min, n = 8). These changes occurred several minutes before the decrease in either SPD-5 or GIP-1 (*Figure 3D–E*). PLK-1 and TAC-1 showed a similar disassembly behavior as SPD-2 and MZT-1, with a gradual decrease in intensity beginning at 2 min post-NEBD (n = 7, *Figure 3—figure supplement 1A*). As expected from our observation of their individual localization behaviors, both SPD-5 and GIP-1 colocalized during the process of disassembly (*Figure 3C*, *Video 5*). Both proteins began a rapid decrease in intensity following their peak at 3 min post-NEBD (SPD-5: 3.00 ± 0.14 min, n = 11; GIP-1: 3.18 ± 0.12 min, n = 11; *Figure 3D–E*). Their volume, however, remained unchanged until 6 min post-NEBD (SPD-5: 5.91 ± 0.17 min, n = 11; GIP-1: 6.00 ± 0.19 min, n = 11), at which time we began to see changes in the structural integrity of the PCM as holes appeared. A qualitative assessment of when these holes began to appear tracked perfectly with the quantitative changes we observed in SPD-5 and GIP-1 PCM volume. We refer to the appearance of these holes and the concomitant change in PCM volume as 'rupture'. Following rup-

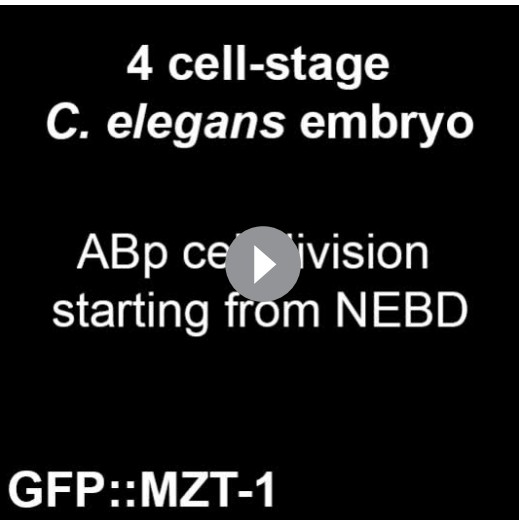

**Video 2.** Centrosome disassembly in the ABp cell in a 4-cell embryo expressing endogenous GFP::MZT-1. Scale bar, 5 μm.
DOI: https://doi.org/10.7554/eLife.47867.008

**Video 3.** Centrosome disassembly in the ABp cell in a 4-cell embryo expressing endogenous GFP::SPD-5. Yellow arrowhead and 'c' mark the centrioles. White arrowhead and 'p' mark the packets. Scale bar, 5 μm.
DOI: https://doi.org/10.7554/eLife.47867.009

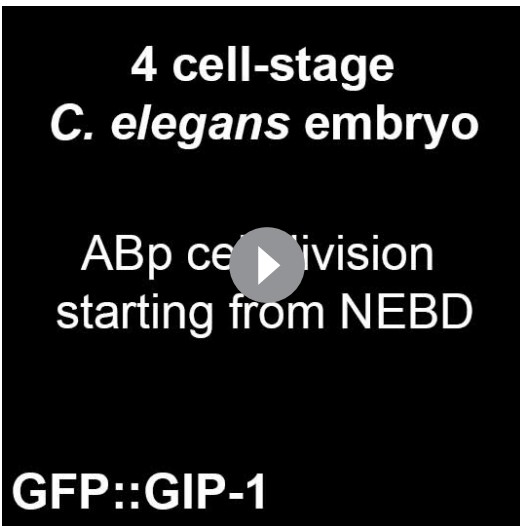

**Video 4.** Centrosome disassembly in the ABp cell in a 4-cell embryo expressing endogenous GFP::GIP-1. Yellow arrowhead and 'c' mark the centrioles. White arrowhead and 'p' mark the packets. Scale bars, 5 μm. DOI: https://doi.org/10.7554/eLife.47867.010

ture, SPD-5 and GIP-1 deformation continued until distinct sub-PCM 'packets' could be observed individualized from the two future centrosomes (*Figure 3A*). Intriguingly, both AIR-1 and TPXL-1 appeared to spread onto the microtubules starting 3 min post-NEBD (*Figure 2C*, *Figure 3—figure supplement 1B*), 3 min ahead of SPD-5 and GIP-1 rupture, before also localizing in packets. Together, these data indicate that the PCM disassembles in two distinct steps: a dissolution step that is characterized by the decrease in intensity of PCM proteins that starts with the removal of the most internal proteins, PLK-1, SPD-2, TAC-1 and MZT-1; and a rupture step where the deformation and subsequent separation of the PCM leads to further disassembly into individual packets.

## Cortical forces mediate the disassembly of the PCM and more specifically SPD-5

The formation of packets that appear to be pulled away from the centrioles suggests that mechanical forces underlie this aspect of PCM disassembly. Forces can be exerted on the PCM by a conserved cortically anchored complex of LIN-5/NuMA, (GPR-1/2)/LGN, and (GOA-1/GPA-16)/Gαi, which localizes dynein-dynactin and can pull on the astral microtubules extending from the PCM (*Kotak and Gönczy, 2013*). Given that greater cortical forces exist in the posterior of the one-cell *C. elegans* embryo, it has been hypothesized that these forces could be responsible for the asynchrony observed in the disassembly of the anterior vs. the posterior centrosome (*Grill et al., 2001*). Moreover, a recent study implicated the (GPR-1/2)/LIN-5/DHC-1 complex in SPD-5 disassembly from the PCM (*Enos et al., 2018*).

To assess the involvement of cortical forces in the general disassembly of the PCM and specifically in rupture and packet formation, we used RNAi to either decrease (*gpr-1/2*(RNAi)) or increase (*csnk-1*(RNAi)) cortical forces. In control embryos treated with lacZ RNAi, SPD-5 ruptured starting 6 min post-NEBD (5.91 ± 0.16 min, n = 11; *Figure 4A*). In contrast, we did not observe SPD-5 rupture or packet formation in *gpr-1/2*(RNAi) treated embryos (*Figure 4A*). Instead, SPD-5, like SPD-2, disassembled by gradual dissolution as indicated by the steady decrease in SPD-5 centrosomal volume which was in sharp contrast to the precipitous drop off seen in control embryos (*Figure 4B*). In *csnk-1*(RNAi) treated embryos, we observed slightly earlier SPD-5 rupture (5.4 + 0.2 min, n = 7; *Figure 4A*). In contrast to SPD-5, SPD-2 disassembly was unaffected following depletion of either *gpr-1/2* or *csnk-1* by RNAi (*Figure 4C*). Interestingly, SPD-5 levels at the PCM were increased by *gpr-1/2* and decreased by *csnk-1* depletion (*Figure 4A*, *Figure 4—figure supplement 1A*). Together, these results suggest that cortical forces generate the mechanical forces necessary for rupture and packet formation, allowing for the efficient removal of the outer sphere protein SPD-5 but not the exclusively inner sphere protein SPD-2.

Cortical forces could be present and constant throughout mitosis or instead intensify at the time of disassembly as is the case in the zygote, providing forces only when necessary (*Gönczy, 2005*; *Rose and Gonczy, 2014*). To distinguish between these possibilities, we tracked the localization of microtubules, LIN-5, DNC-1/dynactin and DHC-1/dynein heavy chain, during different stages of mitosis. Astral microtubules showed a striking network reorganization post-NEBD, growing progressively longer and contacting the cell cortex, sometimes wrapping around the membrane prior to rupture and packet formation (*Figure 4—figure supplement 2A*). We saw a similar reorganization of AIR-1 and TPXL-1, which coat these astral microtubules (*Figure 4—figure supplement 1B*). In contrast, we saw no change in the gross cortical distribution or intensity of LIN-5 (*Figure 4—figure supplement 2B*), DNC-1 (*Figure 4—figure supplement 2C*), or DHC-1 (*Figure 4—figure supplement*

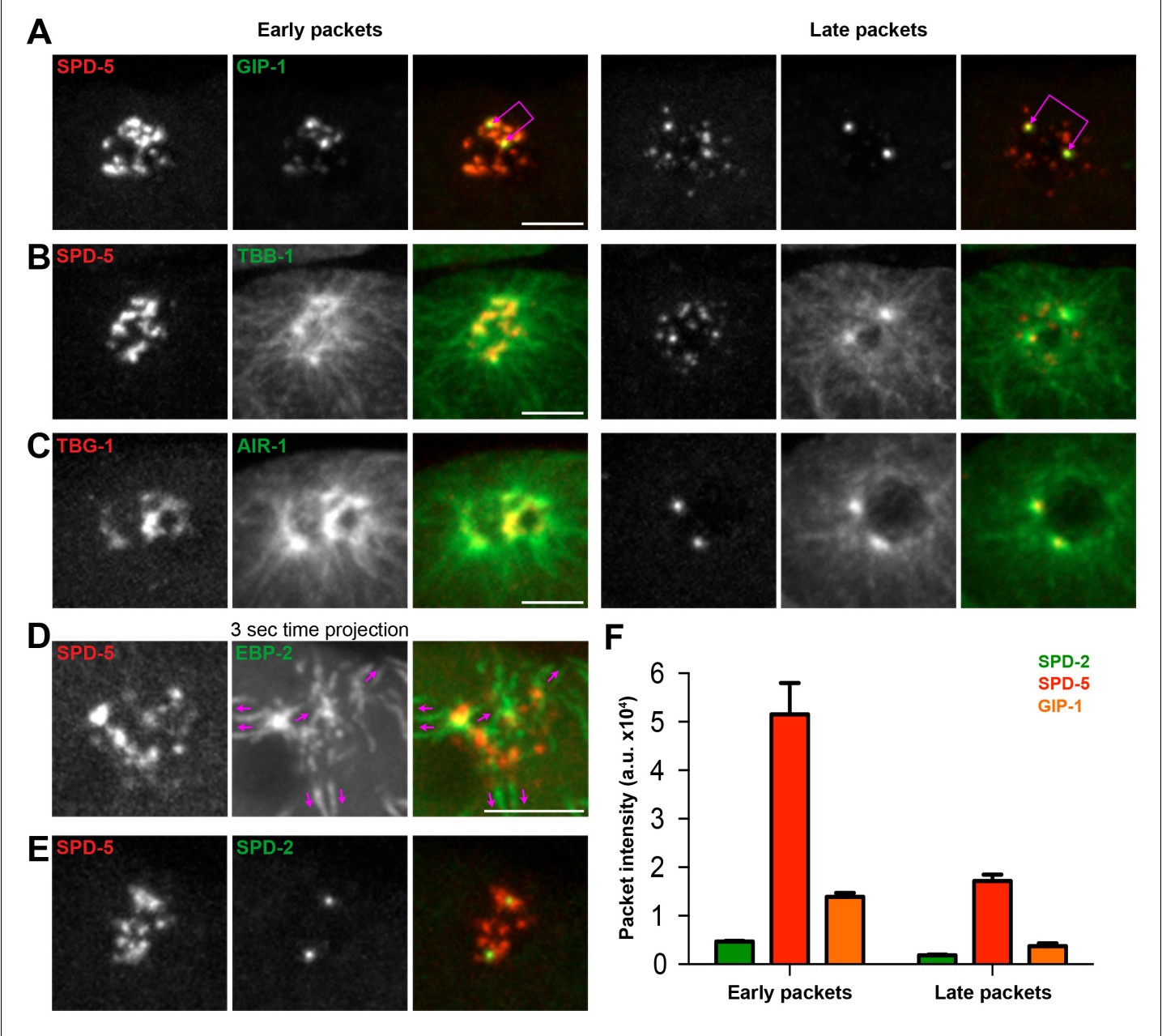

**Figure 2.** The PCM fragments into SPD-5 and GIP-1 containing packets that localize dynamic microtubules. (A–C) Analysis of colocalization of SPD-5 packets (red) with GIP-1 (A, green), or microtubules (B, TBA-1/α-tubulin, green), and TBG-1 (red) with AIR-1(C, green) in early packets (left panels) or late packets (right panels). (C) Three second time projection of EBP-2 (green) showing that packets (SPD-5, red) associate with dynamic microtubules. Magenta arrows represent the direction of EBP-2 movement. Scale bar, 10 μm. (D) Colocalization of SPD-5 packets (red) with SPD-2 (green). Note that SPD-2 does not localize to the packets. (E) Average pixel intensity of SPD-2 (green, n = 8), SPD-5 (red, n = 11), and GIP-1 (orange, n = 8) in early and late packets. 'a.u.'=arbitrary units. Graph represent mean ± s.e.m. Underlying centrioles and corresponding newly forming centrosomes are indicated by magenta joined double arrows.

DOI: https://doi.org/10.7554/eLife.47867.006

2D) post-NEBD. Interestingly, we observed an ephemeral redistribution of DHC-1 coincident with rupture (*Figure 4—figure supplements 2E* and 4 min.). This pattern of localization suggests that although cortical complexes are present throughout the cell cycle, they may only make productive contact with astral microtubules at a particular time period to allow for outer sphere disassembly.

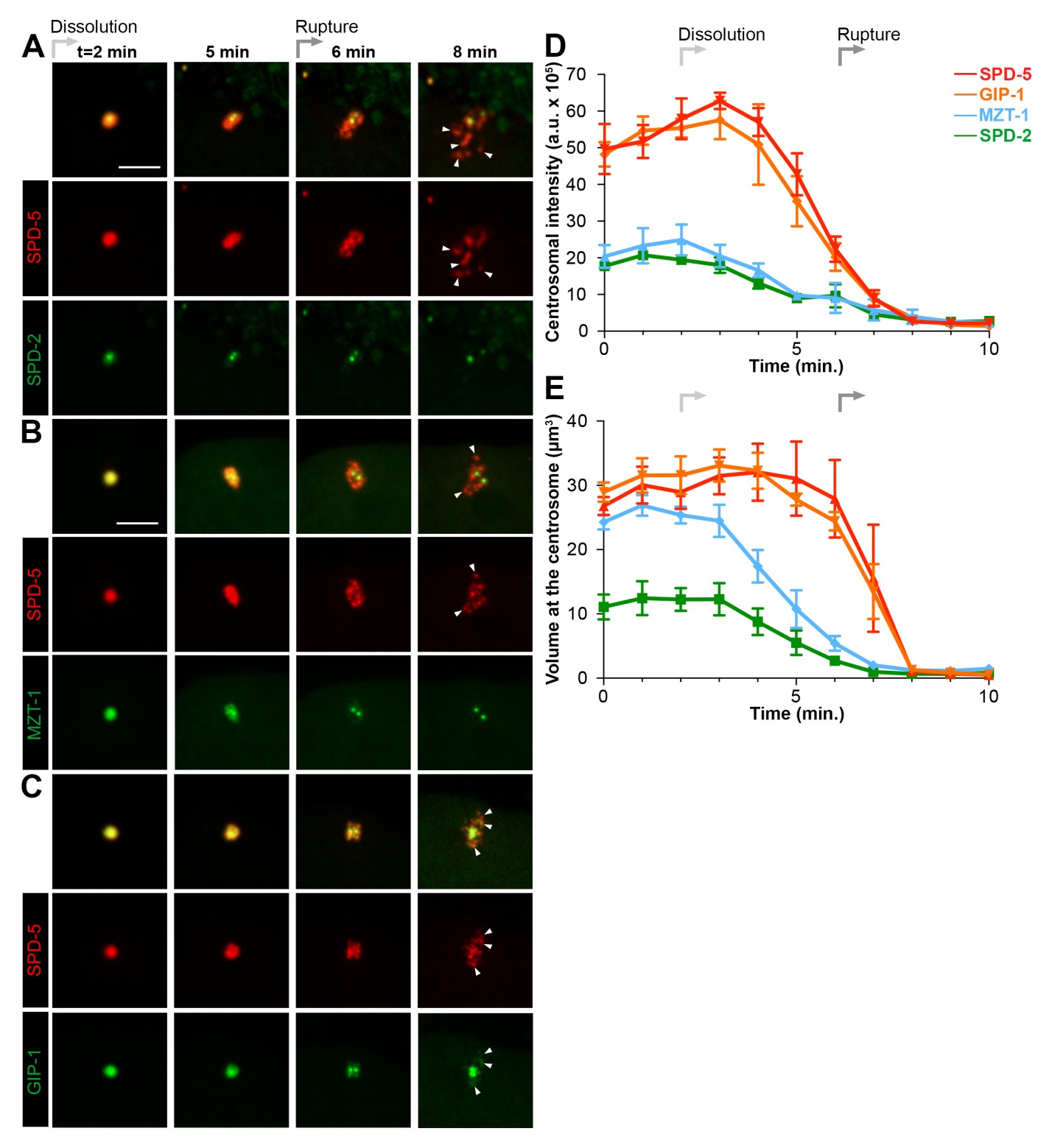

**Figure 3.** Dissolution of SPD-2 and MZT-1 precedes rupture and packet formation, see also *Figure 3—figure supplement 1* and *Video 5*. (A–C) Comparison of tagRFP::SPD-5 (red) to SPD-2::GFP (**A**, green), GFP::MZT-1 (**B**, green), or GFP::GIP-1 (**C**, green) disassembly. 'Dissolution' (light grey arrow) begins as SPD-2 (t = 2 min. post-NEBD) and then MZT (t = 3 min post-NEBD) are removed from the centrosome. 'Rupture' (medium grey arrow) is indicated by holes appearing in the matrix of SPD-5 and GIP-1 surrounding the centrioles, followed by the appearance of individual 'packets' (white arrowheads) of SPD-5 and GIP-1. Scale bar, 10 μm. (**D–E**) Average pixel intensity (**D**) and volume (**E**) at the centrosome of PCM proteins during

*Figure 3 continued on next page*

*Figure 3 continued*

disassembly starting at NEBD (t = 0 min): tagRFP::SPD-5 (red, n = 11), GFP::GIP-1 (orange, n = 9), GFP::MZT-1 (blue, n = 10), SPD-2::GFP (green, n = 8). 'a.u.'=arbitrary units. Graph lines indicate mean ± s.e.m.

DOI: https://doi.org/10.7554/eLife.47867.011

The following figure supplement is available for figure 3:

**Figure supplement 1.** Dynamics of PLK-1, TAC-1, AIR-1 and TPXL-1 during disassembly.

DOI: https://doi.org/10.7554/eLife.47867.012

The rapid rounds of PCM assembly and disassembly during the early embryonic divisions suggest that efficient and robust PCM disassembly might be critical for subsequent carefully timed events such as centriole separation and the assembly of new PCM in the next cell cycle (*Cabral et al., 2013*). We tested whether force dependent PCM removal corresponds to centriolar separation by tracking SAS-4::GFP during disassembly (*Figure 4D*). In control embryos, the centriolar pair appeared as a single SAS-4 focus up to 5 min post-NEBD (*Figure 4D*). Two closely apposed SAS-4 foci became apparent beginning at 5 min post-NEBD (Stage 1, *Figure 4D*), which quickly separated by greater than 1 µm beginning about 1 min later (Stage 2, *Figure 4D*). We saw a significant delay in the onsets of both Stage one and Stage two in *gpr-1/2*(RNAi) treated embryos, but no significant change in *csnk-1*(RNAi) treated embryos (*Figure 4D*). These results suggest that cortical forces facilitate centriole separation either through direct force transmission or indirectly through their role in PCM removal. That *csnk-1* RNAi had no effect on the timing of centriole separation suggests that a force-independent licensing event is necessary to initiate separation (*Cabral et al., 2013*; *Tsou and Stearns, 2006*), but that centrioles are subsequently held together by PCM. In addition to defects in centriole separation, we observed that *gpr-1/2*(RNAi) treated embryos had defects in effectively clearing SPD-5, but not SPD-2, from the PCM prior to the subsequent round of PCM accumulation in the next cell cycle (*Figure 4B and E*). Consistent with these defects, the timing of subsequent SPD-5 accumulation was significantly delayed as compared to control embryos (*Figure 3—figure supplement 1C*). Together, these results underscore the importance of the timely removal of PCM to the developing embryo.

## PP2A phosphatases are required for PCM dissolution

As the growth of the PCM is highly dependent on phosphorylation and CDK inhibition causes precocious removal of PCM proteins (*Woodruff et al., 2014*; *Yang and Feldman, 2015*), we hypothesized that the dissolution of the PCM that precedes rupture and packet formation requires phosphatase activity. To test this hypothesis, we treated cycling embryonic cells at anaphase with either a broad-spectrum serine/threonine phosphatase inhibitor (okadaic acid, OA) or a PP2A inhibitor (rubratoxin A, *Figure 5A*). We observed a stabilization of the PCM in both OA and rubratoxin A treated embryos compared to control embryos treated with DMSO. Notably, treatment with either drug led to depolymerization of the microtubules, perhaps due to the hyperactivation of the depolymerizing kinesin KLP-7 during PP2A inactivation (*Schlaitz et al., 2007*).

Consistent with these pharmacological inhibition results, a recent study implicated the PP2A subunit LET-92 in SPD-5 disassembly (*Enos et al., 2018*). To assess the function of LET-92 on PCM disassembly in general and more specifically on dissolution and packet formation, we treated SPD-2::GFP; tagRFP::SPD-5 expressing embryos with *let-92*(RNAi). As previously reported, *let-92* inhibition caused severe defects in cell division, necessitating analysis in the one-cell zygote rather than 4-cell embryo (*Song et al., 2011*). We monitored PCM disassembly in the one-cell zygote beginning when the membrane invagination that occurs during cytokinetic furrow formation was visible. At this stage in control embryos, PCM disassembly occurs in a similar manner to

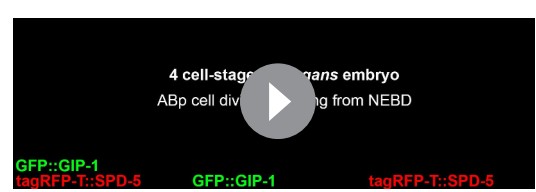

**Video 5.** Centrosome disassembly in the ABp cell in a 4-cell embryo expressing endogenous tagRFP-T::SPD-5; GFP::GIP-1. Yellow arrowhead and 'c' mark the centrioles. White arrowhead and 'p' mark the packets. Scale bars, 5 µm.

DOI: https://doi.org/10.7554/eLife.47867.013

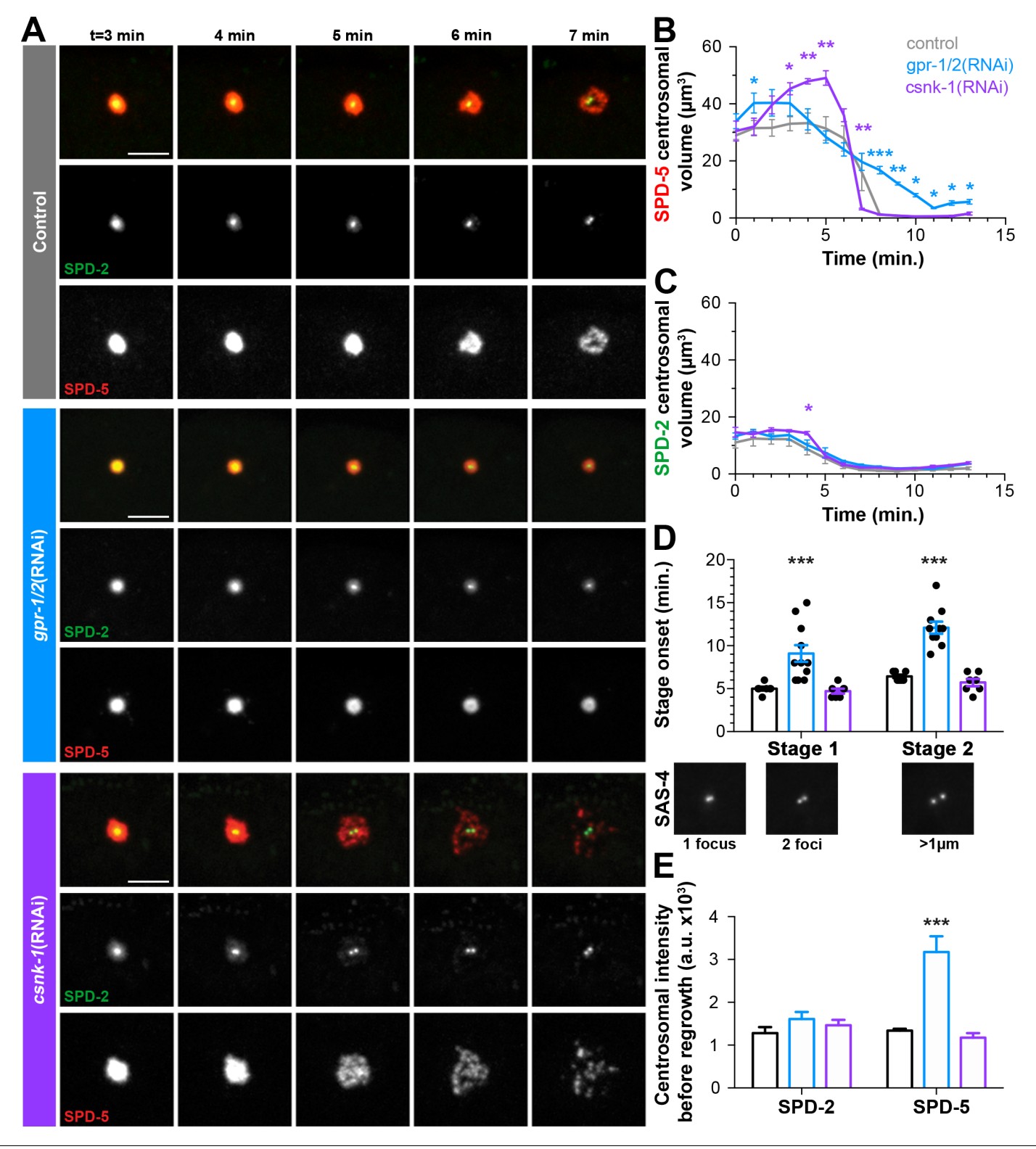

**Figure 4.** Cortical forces rupture the PCM into packets, see also *Figure 4—figure supplement 1* and *Figure 4—figure supplement 2*. (A) Time-lapse analysis starting at NEBD (t = 0 min) of the disassembly of endogenous tagRFP::SPD-5 (red) and SPD-2::GFP (green) treated with lacZ(RNAi) (control, top panels, grey (A–E)), *gpr-1/2*(RNAi) (middle panels, blue (A–E)), or *csnk-1(RNAi)* (bottom panels, purple (A–E)). Scale bar, 10 μm. (B–C) Average volume at the centrosome of SPD-5 (B) or SPD-2 (C) during disassembly starting at NEBD (t = 0 min). (D) Average onset time for centriole separation starting at NEBD (t = 0 min). Stage 1: Centrioles are apparent as a single focus and then double foci of GFP::SAS-4. Stage 2: Centrioles appear >1 μm

*Figure 4 continued on next page*

*Figure 4 continued*

apart. control, Stage 1: 5.00 ± 0.218 min; control, stage 2: 6.429 ± 0.202 min, n = 8; *gpr-1/2*(RNAi), Stage 1: 9.091 ± 0.977 min, *gpr-1/2*(RNAi), Stage 2: 12.100 ± 0.706 min, n = 11; *csnk-1*(RNAi), Stage 1: 4.714 ± 0.286 min, *csnk-1*(RNAi), Stage 2: 5.714 ± 0.421 min, n = 7. (E) Average intensity of SPD-2 or SPD-5 remaining at the centrosome before regrowth in the next cell cycle. SPD-2(control): 1281 ± 139, SPD-5(control): 1337 ± 47, n = 8; SPD-2(*gpr-1/2* (RNAi)): 1610 ± 166, SPD-5(*gpr-1/2*(RNAi)): 3173 ± 369, n = 11; SPD-2(csnk-1(RNAi)): 1467 ± 122, SPD-5(*csnk-1*(RNAi)): 1172 ± 110, n = 7. Asterisks indicate comparison between indicated perturbation and control: *p-value<0.01, ** p-value<0.001, *** p-value<0.0001. 'a.u.'=arbitrary units. Graphs indicate mean ± s.e.m.

DOI: https://doi.org/10.7554/eLife.47867.014

The following figure supplements are available for figure 4:

**Figure supplement 1.** Cortical forces regulate SPD-5, but not SPD-2, intensity and regrowth in the next cell cycle.
DOI: https://doi.org/10.7554/eLife.47867.015

**Figure supplement 2.** Localization of astral microtubules and cortical force generating proteins during PCM disassembly.
DOI: https://doi.org/10.7554/eLife.47867.016

ABp cells, with SPD-2 dissolution preceding SPD-5 rupture and packet formation (*Figure 5B*). *let-92* depletion impaired the disassembly of SPD-2 and SPD-5 from the centrosome in three distinct ways (*Figure 5C*). First, we never saw holes appearing in centrosomal SPD-5, indicating a defect in rupture. Moreover, SPD-5 was only partially cleared into packets, however these packets were more fluid and persisted significantly longer in the cytoplasm than control. Unlike in the 4-cell embryo, we occasionally saw a small fraction of SPD-2 being cleared from the centrosome by rupture in this first cell division, a short-lived phenomenon that was exacerbated when both SPD-2 and SPD-5 were endogenously tagged. In contrast, following *let-92* depletion, SPD-2 consistently ruptured and appeared in packets that persisted in the cytoplasm long after those of control embryos. Second, the rate and time of SPD-2 and SPD-5 disassembly were significantly slower in *let-92* depleted embryos than in control, as indicated by tracking the total centrosomal SPD-2 and SPD-5 over time (*Figure 5E,F*). Centriole duplication fails following *let-92* depletion such that each centrosome at this stage contains only one rather than two centrioles (*Song et al., 2011*). Thus, total centrosome intensity measurements underestimate differences between control and *let-92* depletion conditions because centriole number defects alter the underlying amounts of centriole-localized SPD-2 or SPD-5. Finally, we found that although much of the SPD-2 and SPD-5 appeared to be cleared from the PCM into packets, *let-92* depletion inhibited the complete removal of either protein from the centrosome (*Figure 5C,G*).

The partial removal of SPD-2 and SPD-5 in packets suggested that *let-92* depletion affected mainly dissolution, and that much, but not all, of the remaining PCM was cleared by cortical forces. To test this model, we inhibited *let-92* together with *gpr-1/2* and observed a strong stabilization of both SPD-2 (*Figure 5D,E*) and SPD-5 (*Figure 5D,F*) at the PCM without rupture or packet formation. Together, these results indicate that PP2A phosphatases control the dissolution of SPD-2 and SPD-5, and that both PP2A and cortical forces are required for the efficient and timely removal of the PCM from the centrosome.

## Phosphatases are present and active at the centrosome throughout mitosis

The timing of centrosome disassembly is critical. For example, precocious removal of PCM would impair the ability of the centrosome to build the mitotic spindle, and delayed disassembly affects the subsequent centrosome duplication cycle (see above, *Figure 4*). While cortical pulling forces appear to act on the centrosome post-NEBD (*Figure 4—figure supplement 2A*), it is unclear when phosphatases such as LET-92 are active to help drive disassembly. Phosphatases could be active at the centrosome throughout the cell cycle or could instead be activated only at the time of disassembly. To distinguish between these possibilities, we first assessed the localization of endogenously-tagged LET-92 throughout mitosis. LET-92 localized to the centrosome through the entire process of assembly and disassembly (*Figure 6A*), extending into the outer sphere in a similar localization pattern to SPD-5 and GIP-1 (*Figure 1—figure supplement 1D*). Similarly, LET-92 displayed disassembly behavior and kinetics similar to that of SPD-5 and GIP-1 (*Figure 6A*), however the low expression of LET-92 made it difficult to reliably determine whether it localized to packets.

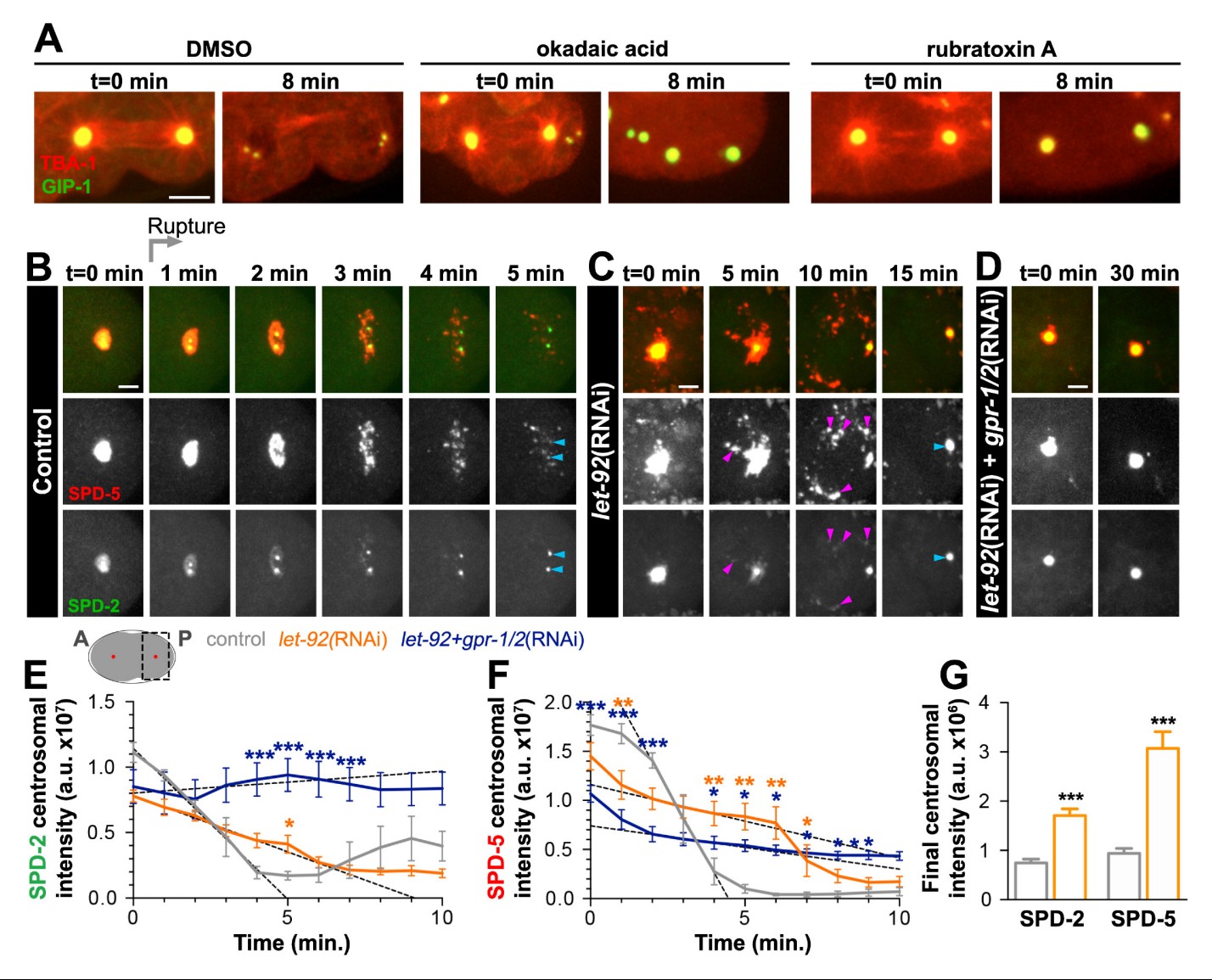

**Figure 5.** PP2A phosphatases regulate PCM disassembly. (A) Time-lapse analysis of embryos expressing *pie-1p*::mCherry::TBA-1/α-tubulin (red) and endogenous GFP::GIP-1 (green) and treated at anaphase (t = 0 min) with DMSO (left panels), 30 μM okadaic acid (middle panels), or 60 μM rubratoxin A (right panels). Scale bar, 10 μm. (B–D) Time-lapse analysis of the disassembly of endogenous tagRFP::SPD-5 (red); SPD-2::GFP (green) starting from cytokinetic furrow ingression (t = 0 min) in the one cell embryo as represented on the cartoon below. Timing of rupture (light gray arrow) at this stage is indicated. Images show posterior (P) embryonic region (black dotted box in cartoon) containing the posterior centrosome (red dot in cartoon). Embryos are treated with lacZ(RNAi) (control, (B), *let-92*(RNAi) (C), or *let-92*(RNAi) +*gpr-1/2*(RNAi) (D). Note the appearance of SPD-2 in packets (C, magenta arrowheads) following *let-92* RNAi treatment. Scale bars, 10 μm. (E–F) SPD-2 (E) or SPD-5 (F) intensity at the centrosome during disassembly starting from cytokinetic furrow ingression (t = 0 min) in embryos treated with lacZ(RNAi) (control, grey, n = 8), *let-92*(RNAi) (orange, n = 8), or *let-92* +*gpr-1/2* (RNAi) (navy, n = 8). SPD-2 disassembly slope (E, 0 to 4 min, black dotted lines): control (slope = $-2.31e^{+6}$, $r^2$ = 0.97), *let-92*(RNAi) (slope = $-8.60e^{+5}$, $r^2$ = 0.94) and *let-92* +*gpr-1/2*(RNAi) (slope = $1.67e^{+5}$, $r^2$ = 0.86). SPD-5 disassembly slope (F, 2 to 4 min, black dotted lines): control (slope = $-5.65e^{+6}$, $r^2$ = 0.95), *let-92*(RNAi) (slope = $-7.46e^{+5}$, $r^2$ = 0.92) and *let-92* +*gpr-1/2*(RNAi) (slope = $-4.40e^{+5}$, $r^2$ = 0.91). Slopes are significantly different from each other (t-test, p-value<0.0001). (G) Average centrosomal pixel intensity at the end of disassembly in control (t = 5', grey, n = 15) and in *let-92*(RNAi) treated embryos (t = 15', orange, n = 13). Note that we accounted for centriole duplication defects following *let-92* depletion by comparing the average intensity of each individual centriole/centrosome in control embryos (see two SPD-2 foci representing two individual centrioles/centrosomes, light blue arrowheads at t = 5' in *Figure 5B*) to intensity of the single centrosome in *let-92* depleted embryos (single SPD-2 focus, light blue arrowhead at t = 15' in *Figure 5C*; see Material and methods). Asterisks indicate comparison between indicated perturbation and control: *p-value<0.01, ** p-value<0.001, *** p-value<0.0001. 'a.u.'=arbitrary units. Graphs indicate mean ± s.e.m.

DOI: https://doi.org/10.7554/eLife.47867.017

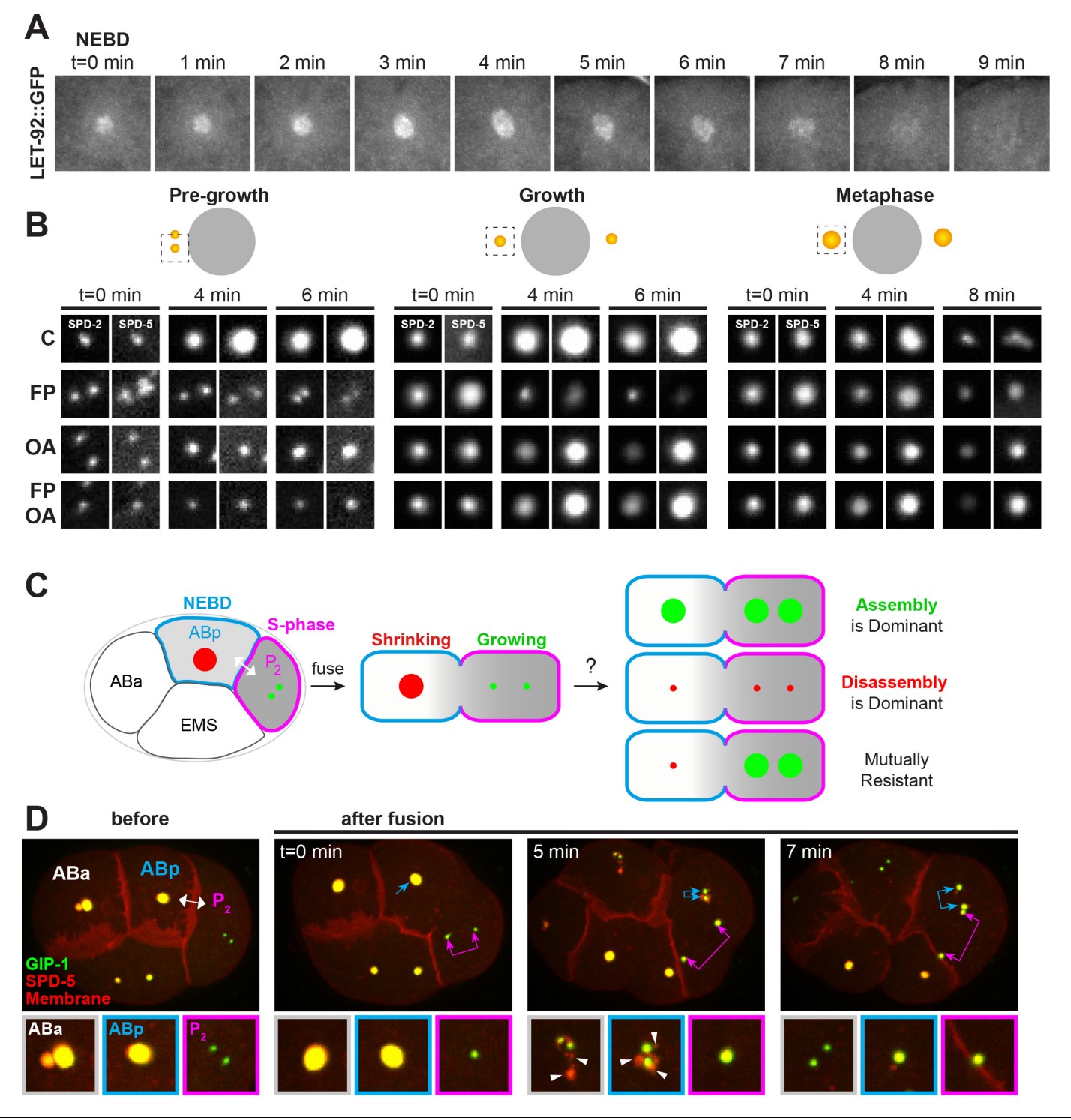

**Figure 6.** Kinases and phosphatases shape the PCM throughout mitosis, see also *Figure 6—figure supplement 1*. (A) Time-lapse analysis of the disassembly of endogenous LET-92 starting at NEBD (t = 0 min) and imaged every minute for 9 min. (B) Time-lapse analysis of embryos expressing endogenous tagRFP-T::SPD-5 (red) and SPD-2::GFP (green) and treated at pre-growth (t = 0 min, *left panels*), growth (t = 0 min, *middle panels*) or metaphase (t = 0 min, right panels) with DMSO ('C', first row), 200 μM flavopiridol ('FP', second row), 30 μM okadaic acid ('OA', third row), or flavopiridol and okadaic acid ('FP, OA', fourth row). (C) Cartoon showing possible outcomes from the cell fusion experiment of a post-NEBD mitotic embryonic cell (ABp, blue) with decreasing levels of kinases (grey) and a S-phase embryonic cell, ($P_2$, magenta) with high levels of kinases. Assembling (green) or disassembling (red) centrosomes are depicted. (D) Time-lapse analysis of the ABp – $P_2$ fusion experiment in embryos expressing endogenous

*Figure 6 continued on next page*

*Figure 6 continued*
tagRFP-T::SPD-5 (red) and GFP::GIP-1 (green) and overexpressing PLCδ::mCherry (red). Fusion site is marked by the double-headed white arrow. Top images show the entire embryo and bottom images show a magnification of the control Aba centrosome (white), the disassembling ABp centrosome (blue), and the assembling $P_2$ centrosome (magenta). Packets are marked with white arrowheads.
DOI: https://doi.org/10.7554/eLife.47867.018
The following figure supplement is available for figure 6:

**Figure supplement 1.** Microtubule behavior following cell fusion.
DOI: https://doi.org/10.7554/eLife.47867.019

These localization data raised the possibility that phosphatases are active at the centrosome throughout mitosis rather than just during the disassembly phase. To test this possibility further, we treated cycling SPD-2::GFP; tagRFP::SPD-5 embryos with OA and/or the CDK inhibitor flavopiridol (FP) at three defined timepoints during mitosis (*Figure 6B*): (1) just prior to PCM growth ('pre-growth'); (2) during PCM growth ('growth'); and (3) at metaphase immediately before NEBD when PCM levels at the centrosome are near their peak and just prior to the initiation of disassembly ('metaphase'). FP treatment in pre-growth cells inhibited the accumulation of SPD-2 and SPD-5 at the centrosome and growth-stage FP treatment induced their precocious disassembly. Metaphase stage FP treatment had relatively little effect on SPD-2 and SPD-5 centrosomal localization, consistent with the fact that CDK is normally inactivated shortly after metaphase (*Kipreos and van den Heuvel, 2019*). SPD-2 and SPD-5 both accumulated at centrosomes following pre-growth OA treatment, albeit to a lesser extent than in control cells. As predicted by the *let-92* RNAi phenotype (*Figure 5B–D*), growth and metaphase stage PCM was stabilized by OA treatment as indicated by the continued presence of SPD-5 (*Figure 6B*) or GIP-1 (*Figure 5A*) at the centrosome during the disassembly period. Surprisingly and in contrast to SPD-5, SPD-2 was precociously disassembled in the presence of OA in growth and metaphase stage embryos.

The precocious disassembly of PCM following CDK inhibition suggested that the association of PCM proteins with the centrosome is normally actively opposed such that turning off assembly immediately triggers disassembly. To test if this opposition is phosphatase dependent, we treated embryos with both FP and OA. Pre-growth stage treated embryos showed no addition of SPD-2 or SPD-5, consistent with a requirement for CDK activity in centrosome maturation. Treatment with both inhibitors at growth or metaphase stage led to a stabilization of SPD-5, consistent with the hypothesis that SPD-5 assembly driven by CDK activity is normally opposed by phosphatase activity. In contrast, SPD-2 was precociously disassembled in the presence of OA and FP in growth and metaphase stage embryos, identically to what was observed in embryos treated with OA alone. These results suggest that the maintenance of SPD-2 at the centrosome is controlled by an OA sensitive phosphatase and indicate that regulation of SPD-2 and SPD-5 can be uncoupled.

## Centrosome assembly and disassembly are mutually resistant processes

The differential behavior of PCM proteins in response to the timing of kinase and phosphatase inhibition suggests that assembling and disassembling PCM are inherently different structures. We therefore wanted to test whether the factors that contribute to PCM assembly had any impact on disassembling PCM or vice versa. Using in vivo cell fusion experiments, we previously found that cytoplasm from pre-anaphase cells could rapidly induce the assembly of PCM and microtubules onto inactive centrosomes in both cycling and differentiated cells, indicating that mitotic cytoplasm dominantly selects for PCM assembly (*Yang and Feldman, 2015*). Using a similar cell fusion approach, we fused a pre-metaphase cell in which the PCM was assembling ($P_2$, *Figure 6C*) and a post-anaphase cell in which the PCM was disassembling (ABp, *Figure 6C*) to examine the relationship between assembling and disassembling PCM and the cytoplasmic environments that maintain them. If PCM assembly is dominant, we would expect the PCM in ABp to be stabilized following cell fusion. Conversely, if PCM disassembly is dominant, we would expect cell fusion to induce disassembly of the $P_2$ centrosome. Finally, the process of assembly and disassembly could be mutually resistant to the factors that induce the converse process, that is cell fusion would have no impact on the assembling $P_2$ centrosome or the disassembling ABp centrosome.

We fused ABp (*Figure 6D*, blue, n = 13) with P$_2$ (*Figure 6D*, magenta) 2 min after NEBD in the ABp cell in embryos expressing a membrane localized mCherry and endogenously tagged tagRFP::SPD-5 and GFP::GIP-1. Microtubules associated with the disassembling ABp centrosome invaded P$_2$ following fusion, confirming an exchange between the cytoplasm of the two cells (*Figure 6—figure supplement 1A*). Following fusion, the ABp centrosome (*Figure 6D*, blue arrows) exhibited normal disassembly, showing packet formation as in the control unfused ABa cell (*Figures 6D* and *5 min*, white arrowheads). Similarly, the P$_2$ centrosome showed normal assembly following cell fusion (*Figure 6D*, magenta arrows). Interestingly, as soon as the existing PCM was stripped from the ABp centrosome into packets, new PCM rapidly assembled at the ABp centrosome (*Figures 6D* and *7 min*, blue double arrow) in a similar manner to that of P$_2$ (*Figures 6D* and *7 min*, pink double arrow). The addition of PCM in ABp was precocious as the control ABa cell had not yet started adding PCM to its centrosome. This precocious assembly also induced the assembly of microtubules but did not lead to the clustering of the ABp and P$_2$ centrosome (*Figure 6—figure supplement 1A*). Together, these results indicate that PCM assembly and disassembly are mutually resistant with each state being locked in place; a disassembling centrosome and PCM packets are unaffected by cytoplasm that normally promotes assembly and an assembling centrosome is unaffected by cytoplasm that promotes disassembly. Moreover, the addition of new 'assembly state' PCM occurs once the old 'disassembly state' PCM is removed. Previous experiments indicated that fusion induced PCM assembly requires CDK activity (*Yang and Feldman, 2015*), lending further evidence to the idea that the nature of the PCM changes throughout mitosis and becomes resistant to phosphoregulation.

## Discussion

Here we present evidence that MTOC function at the centrosome is inactivated through a two-step PCM disassembly process involving the gradual dissolution of proteins localized close to the centrioles followed by the forceful rupture and ejection of proteins that extend more

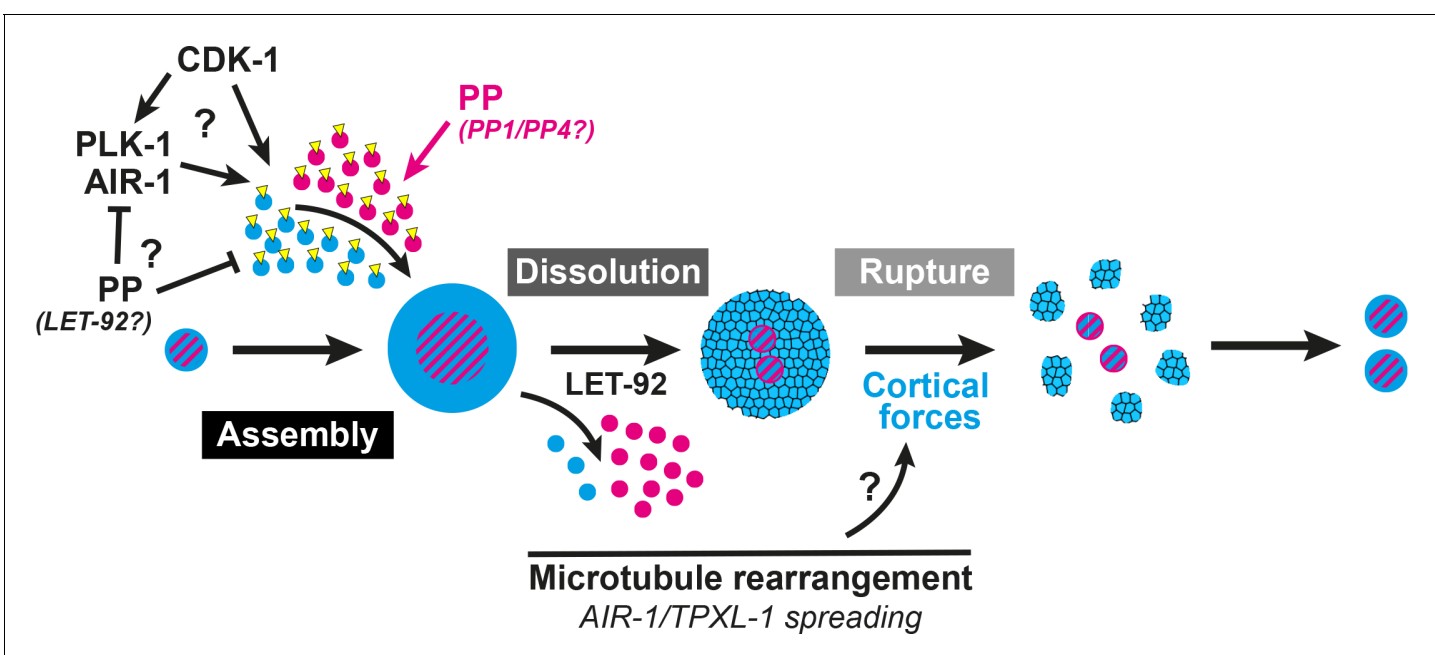

**Figure 7.** A two-step model of PCM disassembly. The centrosome is assembled through the activity of mitotic kinases that phosphorylate PCM proteins to be incorporated into an inner (magenta) and outer (blue) sphere. Phosphatases oppose this process, hypothetically by inactivating kinases and/or directly dephosphorylating PCM proteins thereby promoting their disassembly. As kinase activity is naturally attenuated in the cell cycle, phosphatase/LET-92 activity dominates, resulting in PCM dissolution. Microtubules lengthen and more readily contact the cortex with TPXL-1/AIR-1 spreading along those microtubules. This rearrangement could be a key aspect in the regulation of cortical forces that ultimately rupture an aging outer sphere of PCM proteins (blue and black lattice) into packets.
DOI: https://doi.org/10.7554/eLife.47867.020

distally. Our data suggest that PCM dissolution is controlled by phosphatase activity, including that of PP2A, and that cortical forces drive the rupture of remaining PCM, pulling it into packets (*Figure 7*). While previous studies indicated a role for both LET-92 and cortical forces in the disassembly of SPD-5 (*Enos et al., 2018*), here we have presented a more complete picture of centrosome disassembly in two steps as discussed below. Furthermore, our data indicate that PCM is not a mass of proteins that is assembled and disassembled as a batch by common regulation, but rather a complicated meshwork of proteins with distinct mechanisms of intricate regulation.

Our two-step model for PCM disassembly is predicated on the duality of localization patterns and disassembly behaviors we discovered for different PCM proteins. In particular, we found that the *C. elegans* centrosome is organized into discrete layers which we propose to be part of two spheres based on the localization boundaries of the matrix proteins SPD-2 and SPD-5. While our analysis of PCM composition is limited by our choice of diffraction-limited imaging platforms, this layered organization appears to be generally conserved between direct and functional orthologs in *C. elegans*, *Drosophila*, and human PCM, suggesting evolutionary pressure to create specific functional PCM domains and that the mechanisms of disassembly described here might be generally conserved. We found that known binding partners separate between these two distinct PCM regions. For example, SPD-5 and GIP-1 localize to the outer sphere region which lacks binding partner SPD-2 and MZT-1, respectively. Similarly, we found that both SPD-2 and MZT-1 normally disassemble from the centrosome before either SPD-5 or GIP-1. Furthermore, the precocious removal of SPD-2 by OA treatment did not affect SPD-5 or GIP-1 localization, suggesting that these proteins have the ability to form a matrix in the absence of SPD-2 and MZT-1. SPD-5 can form a matrix in vitro and perhaps its self-association drives outer sphere assembly and maintains PCM structure in the absence of SPD-2 (*Woodruff et al., 2015*). Finally, the differential localization patterns of PCM proteins correlate with the two different disassembly modes we observed (dissolution vs. rupture): Proximal proteins (PLK-1, SPD-2, MZT-1, TAC-1) disassembled by gradual dissolution while more distally extending proteins (ZYG-9, GIP-1, SPD-5, AIR-1) ruptured and formed packets. These differences in disassembly behaviors might reflect differences in diffusion of individual components as SPD-2 and PLK-1 are known to have increased mobility within the PCM as compared to SPD-5 and GIP-1 (*Laos et al., 2015*; *Woodruff et al., 2017*). Thus, removal of more fluid inner sphere proteins could rely on active turnover while disassembly of more stable outer sphere proteins might require physical disruption.

Our data suggest that PCM disassembly is initiated through the active turnover of inner sphere proteins by dephosphorylation, either through the direct action of phosphatases on these proteins or through their inactivation of mitotic kinases. Indeed, the removal of both SPD-2 and MZT-1 appears to exclusively depend on phosphatase activity as they do not localize in packets and their disassembly was not affected by the inhibition of cortical forces. Furthermore, a pool of both SPD-2 and SPD-5 remained at the centrosome following LET-92 depletion, suggesting that cortical forces alone are not sufficient for their effective clearance. Thus, PCM disassembly appears to be initiated by dephosphorylation by the PP2A subunit LET-92. As LET-92 plays a number of roles at the centrosome and phosphatase activity can directly regulate mitotic kinases (*Enos et al., 2018*; *Kitagawa et al., 2011*; *Song et al., 2011*), further studies will be necessary to determine if its role in PCM dissolution is direct or indirect.

Our inhibitor experiments also uncovered key roles for phosphatases in both centrosome assembly and disassembly. Centrosome assembly appears to be the result of a balance of kinase and phosphatase activity acting on PCM proteins. Both TBG-1 and SPD-5 could be prematurely forced from the centrosome by dissolution in the presence of the CDK inhibitor FP (this study and *Yang and Feldman, 2015*). This precocious dissolution was inhibited by additional treatment with OA, suggesting that the ability of CDK to drive the addition of PCM proteins is actively opposed by serine/threonine phosphatase activity. This phosphatase-based opposition is independent of inhibition of CDK activity and might instead act on other kinases such as PLK-1/PLK1 or AIR-1/Aurora A. Consistently, PP2A can remove activating phosphates from both PLK1 and Aurora A (*Horn et al., 2007*; *Wang et al., 2015*). Alternatively, our data are also consistent with phosphatases being directly inhibited by CDK, thus forced CDK inactivation would relieve phosphatase inhibition and result in dissolution. However, we favor a model in which phosphatases actively oppose PCM assembly as

this type of model can account for the observed mobility of exclusively inner sphere proteins such as SPD-2 (*Laos et al., 2015*). Indeed, phosphatases might directly remove PCM protein phosphorylation which in turn could lead to their dissociation from the centrosome. This model seems plausible as SPD-5 can be dephosphorylated in vitro by LET-92 and has been shown to interact with the PP2A targeting subunits RSA-1 and RSA-2 (*Enos et al., 2018*; *Schlaitz et al., 2007*). As LET-92 is localized to the centrosome throughout mitosis, PCM dissolution might simply be the result of the normal inhibition of CDK activity in the cell cycle coupled with the continued presence of LET-92 at the centrosome.

These experiments have also revealed differential regulation for SPD-2 and SPD-5. In contrast to LET-92 inhibition which stabilized both SPD-2 and SPD-5 at the centrosome, we found that OA treatment inhibited the removal of SPD-5 from the centrosome but expedited SPD-2 disassembly. These experiments suggest that SPD-2 association with the centrosome is positively regulated by another OA sensitive phosphatase, further separating the localization and regulation of SPD-2 from that of SPD-5. OA induced SPD-2 removal only occurred after NEBD, suggesting that the phosphatase that maintains it at the centrosome is regulated in time and/or space. PP1 and/or PP4 could play this role as both have been shown to positively regulate PCM association with the centrosome, and PP1 can interact directly with the SPD-2 homologue CEP192 (*Martin-Granados et al., 2008*; *Nasa et al., 2017*). Thus, PCM disassembly and assembly are regulated by phosphatases and SPD-2 appears to have additional levels of dephosphorylation dependent mechanisms to maintain it the centrosome. In the future, it will be interesting to determine if other inner sphere proteins have similar regulation.

The sensitivity of PCM to kinase and phosphatase activity appears to change during mitosis. Indeed, disassembling PCM appears to be resistant to assembly-competent cytoplasm; PCM continued to disassemble into packets despite exposure to active mitotic kinases following cell fusion. Moreover, the aging PCM matrix appeared to protect centrosomes from the addition of new PCM, which was only added onto centrioles once existing PCM had been stripped away. Likewise, assembling PCM was unaffected by the presence of disassembly-competent cytoplasm following fusion, further underscoring that disassembly by phosphatases is a normal aspect of assembly that is dominated by kinase activity. An alternative explanation for these observed phenomena is that PCM assembly is slower than the off rate of PCM proteins or that diffusion between ABp and $P_2$ is too slow to induce assembly prior to disassembly. However, we previously found that new PCM can add onto inactive centrosomes in interphase cells in under three minutes following fusion to a mitotic cell (*Yang and Feldman, 2015*). Given that in the present study we observe disassembly and subsequent reassembly in ABp well after this three-minute window, our results instead favor the model that centrosomes become locked in mutually resistant assembly or disassembly states perhaps due to a change in the biophysical nature of the PCM. Recent studies of in vitro assembled PCM point to different physical properties between 'young' and 'old' condensates of SPD-5, with young condensates behaving more like a liquid and old condensates acting more like a gel (*Woodruff et al., 2017*).

This change in the nature of the PCM could also be regulated by phosphorylation. For example, Cnn is proposed to live in different states in the PCM in *Drosophila*, assembling first near the centrioles in a phosphorylated state and transiting towards the PCM periphery as a higher order multimerized scaffold where Cnn molecules are likely eventually dephosphorylated and lose PCM association (*Conduit et al., 2014*). Similarly, the inner sphere of SPD-5 may represent a specific pool that can be readily dissociated by dephosphorylation, while the outer sphere may represent a macromolecular scaffold that relies on physical disruption for disassembly. More mobile inner sphere proteins such as SPD-2 could be more likely to escape an aging outer sphere matrix of SPD-5 and other proteins which would then be torn apart by cortical forces as it matured into a gel. Indeed, *let-92* depletion inhibited rupture and led to the appearance of more fluid packets, consistent with a role for phosphorylation in regulating the nature of the PCM. Interestingly, a recent study reported similar defects upon depletion of PCMD-1 (*Erpf et al., 2019*). Thus, PCMD-1 could be involved in this phosphorylation dependent regulation of PCM structural integrity. Different landscapes of phosphorylation could be provided by the complementary localization patterns of the two mitotic kinases PLK-1 and AIR-1. Although previous studies had suggested that only more proximally localized AIR-1 is activated by auto-phosphorylation (*Hannak et al., 2001*; *Toya et al., 2011*), more recent studies have found that human Aurora A can also be activated by interaction with TPX2 (*Zorba et al., 2014*). These results suggest that AIR-1 might be similarly active throughout the outer sphere where

it colocalizes with TPXL-1 and therefore could phosphorylate a complementary set of PCM proteins to that of PLK-1.

Following dissolution, we found that the PCM fragments into small packets that retain MTOC potential. These packets are reminiscent of PCM flares described in *Drosophila* (*Megraw et al., 2002*) and to the fragments that are released from the centrosome in anaphase in the LLC-PK1 kidney cell line (*Rusan and Wadsworth, 2005*). PCM flares are reported to be present to some extent throughout the cell cycle rather than exclusively during centrosome disassembly like the packets we describe (*Lerit et al., 2015*; *Megraw et al., 2002*). However, like packets, flare activity dramatically increases in telophase and centrosome fragments in LLC-PK1 cells appear in anaphase. Flares were first defined by their association with Cnn, the proposed functional ortholog of SPD-5 (*Megraw et al., 2002*). However, flares also localize D-TACC while the *C. elegans* orthologue TAC-1 does not localize to packets. Furthermore, γ-TuRC does not localize to flares but does localize to both packets and centrosome fragments. Finally, packets, flares, and centrosome fragments all appear to be dependent on microtubules for their formation. Thus, these remnants of PCM fragmentation appear to be conserved, although the molecular composition and timing of appearance of the resulting structures can vary. Because packets still disassemble following *let-92* inhibition and exposure to assembly competent cytoplasm, other kinase- and phosphatase-independent mechanisms must be required for packet disassembly. These mechanisms could include rapid diffusion or proteasome-based degradation and further studies will be required to uncover their mechanism of disassembly.

Finally, our results indicate that cortical forces can shape the PCM mainly through an effect on outer sphere proteins. The balance of cortical forces appears to tune the levels of SPD-5 incorporation into the PCM, independently of SPD-2; decreasing or increasing cortical forces caused more or less SPD-5 incorporation, respectively, but had no effect on the levels of SPD-2. Thus, cortical forces negatively regulate the growth of the PCM, hypothetically by physically changing the nature of PCM in the outer sphere. We found that the effect of cortical forces on the PCM was temporally restricted, with the PCM only becoming sensitive to these forces in late anaphase. This claim is supported by multiple observations. First, when PCM is precociously removed from the centrosome by FP treatment, SPD-5 and γ-TuRC disassemble by dissolution, suggesting that cortical forces are not capable of rupturing the PCM and forming packets at the time of normal PCM growth (this study and *Yang and Feldman, 2015*). Second, we see astral microtubule rearrangements starting in anaphase that result in a large increase in the number of microtubules that reach the plasma membrane, consistent with what has been seen in other cell types (*Rusan and Wadsworth, 2005*). Thus, productive force can only act on the PCM beginning in anaphase. Consistently, increasing cortical forces by CSNK-1 inhibition only slightly expedited PCM disassembly. Finally, we see an apparent movement of AIR-1 and TPXL-1 from the PCM along the microtubules also beginning at about the end of anaphase. This redistribution could simply be a biproduct of the microtubule network reorganization. Alternatively, AIR-1 and TPXL-1 relocalization could contribute to microtubule reorganization by stabilizing the microtubules directly or promoting their efficient outgrowth (*Bayliss et al., 2003*; *Zhang et al., 2017*).

In total, these results suggest that PCM is disassembled through the removal of the inner sphere of PCM by phosphatase activity, including that of PP2A. This dissolution is followed by the clearance of an aging outer sphere matrix by cortical pulling forces, which liberate dynamic microtubules and inactivate MTOC function at the centrosome (*Figure 7*). With an understanding of the mechanisms underlying this process, future studies will reveal whether hyperactive MTOC function at the centrosome has a direct effect on the cell cycle or cell differentiation in a developing organism, as has been previously postulated.

## Materials and methods

### *C.elegans* strains and maintenance

*C. elegans* strains were maintained at 20°C unless otherwise specified and cultured as previously described (*Brenner, 1974*). Experiments were performed using embryos from one-day adults. Unless otherwise indicated, at least five embryos were scored in each experimental condition. Strains used in this study are as follows.

| Strain name | Genotype | Source |
|---|---|---|
| N2 | Bristol N2 | CGC |
| JLF14 | gip-1(wow3[gfp::gip-1]) III | (Sallee et al., 2018) |
| JLF432 | spd-2(wow60[spd-2::gfp3xflag]) I | This study |
| JLF359 | spd-5(wow36[tagrfp-t3xmyc::spd-5]) I | This study |
| JLF361 | spd-5(wow52[gfp3xflag::spd-5]) I | This study |
| JLF342 | zif-1(gk117); mzt-1(wow51[gfp3xflag::mzt-1]) I | (Sallee et al., 2018) |
| JLF198 | Zif-1 (gk117); sas-4(wow32[zfgfp3xflag::sas-4]) III | This study |
| JLF50 | zif-1(gk117), outcrossed 6x | (Sallee et al., 2018) |
| JLF427 | spd-5(wow36[tagrfp-t3xmyc::spd-5]) I; unc-119(ed3); ruIs57 [pie-1p::GFP::tbb/β-tubulin; unc-119(+)] | This study/CGC |
| JLF428 | spd-5(wow36[tagrfp-t3xmyc::spd-5]) I; ebp-2(wow47[ebp-2:: gfp3xflag]) II | This study/ (Sallee et al., 2018) |
| JLF430 | spd-5(wow36[tagrfp-t3xmyc::spd-5]) I; gip-1(wow3[gfp3xflag::gip-1]) III | This study/ (Sallee et al., 2018) |
| JLF426 | spd-5(wow36[tagrfp-t3xmyc::spd-5]) I; mzt-1(wow51[gfp3xflag::mzt-1]) I | This study |
| JLF425 | spd-5(wow36[tagrfp-t3xmyc::spd-5]) I; spd-2(wow60[spd-2:: gfp3xflag]) I | This study |
| JLF429 | zif-1(gk117); spd-5(wow36[tagrfp-t3xmyc::spd-5]) I; sas-4(wow32[zfgfp3xflag::sas-4]) III | This study |
| LP585 | lin-5(cp288[lin-5::mNG-C13xFlag]) II | CGC |
| LP560 | dhc-1(cp268[dhc-1::mNG-C13xFlag]) I | CGC |
| LP563 | dnc-1(cp271[dnc-1::mNG-C13xFlag]) I | CGC |
| OD2425 | plk-1(it17[plk-1::sgfp]loxp) III | (Martino et al., 2017) |
| JLF158 | tac-1(wow19[tac::zfgfp3xflag]); zif-1(gk117) | This study |
| JLF105 | zyg-9(wow12[zf::gfp::zyg-9]) II; zif-1(gk117) | (Sallee et al., 2018) |
| JLF518 | let-92(wow88[let-92::gfpaid3xFlag]) IV/nT1 | This study |
| JLF216 | tpxl-1(wow34[zfgfp3xflag::tpxl-1) I; zif-1(gk117) | (Sallee et al., 2018) |
| JLF166 | itSi569(tbg-1::mcherry); air-1(wow14[air-1::zfgfp3xflag]) V; zif-1(gk117) | (Sallee et al., 2018) |
| JLF517 | gip-1(wow3[gfp3xflag::gip-1]) IIIGIP-1,; spd-5(wow36[tagrfp-t3xmyc::spd-5]) I; ltIs44 [pie-1p::mCherry:: PH(PLC1delta1)+unc-119(+)] V | This study/ (Sallee et al., 2018) |
| JLF8 | ruIs75(tubulin::gfp); itIs37 [pie-1p::mCherry::H2B::pie-1 3'UTR + unc-119(+)] IV; ltIs44 [pie-1p::mCherry:: PH(PLC1delta1)+unc-119(+)] V | This study |

## CRISPR/Cas9

Endogenously tagged proteins used in this study were generated using the CRISPR Self Excising Cassette (SEC) method that has been previously described (Dickinson et al., 2015). DNA mixtures (sgRNA and Cas9 containing plasmid and repair template) were injected into young adults, and CRISPR edited worms were selected by treatment with hygromycin followed by visual inspection for appropriate expression and localization (Dickinson et al., 2015). sgRNA and homology arm sequences used to generate lines are as follows:

| Allele | sgRNA sequence | Homology arm | SEC used |
|---|---|---|---|
| spd-2 (wow60[spd-2::gfp3xflag]) | cagagaatatttggaaagttagg (pJM31) | HA1 Fwd: ttgtaaaacgacggccagtcgccggca GTGTTGACATTCGCATCGAC | pDD282 |
| | | HA1 Rev: CATCGATGCTCCTGAGGC TCCCGAT GCTCCCTTTCTATTCGAAAATC TTGTATTGG | |
| | | HA2 Fwd: CGTGATTACAAGGATGACGA TGACAAGAGATAA aatcttaagataactttccaaatattc | |
| | | HA2 Rev: ggaaacagctatgaccatgttatcg atttcatcctcaatatgccagatgc | |
| spd-5 (wow36[tagrfp-t3xmyc::spd-5]) | gaaaacttcgcgttaaATGGAGG (pJM13) | HA1 Fwd: cacgacgttgtaaaacgacggccagtc gacgcaaggaaatcgtcactt | pDD286 |
| | | HA1 Rev: CTTGATGAGCTCCTCTCCC TTGGAGACCATtt aacgcgaagttttctg | |
| | | HA2 Fwd: GAGCAGAAGTTGA TCAGCGAGGAAGA CTTGGAGGATAATTCTGTGC TCAACG | |
| | | HA2 Rev: tcacacaggaaacagctatgaccatgttat CTTTCCTCCATTGCATGCTT | |
| spd-5 (wow52[gfp3xflag::spd-5]) | | HA1 Fwd: acgttgtaaaacgacggccagtcgc cggcaacgcaaggaaatcgtcactt | pDD282 |
| | | HA1 Rev: TCCAGTGAACAATTCTTCTCC TTTACTCAT ttaacgcgaagttttctg | |
| | | HA2 Fwd: CGTGATTACAAGGATGACGA TGACAAGA GAGAGGATAATTCTGTGC TCAACG | |
| | | HA2 Rev: tcacacaggaaacagctatgaccatgttat CTTTCCTCCATTGCATGCTT | |
| tac-1 (wow19[tac-1::zfgfp3xflag]) | cagagaatatttggaaagttagg (pJF283) | HA1 Fwd: ttgtaaaacgacggccagtcgccg gcagctttctaggccaactgcac | pJF250 |
| | | HA1 Rev: ACAAAGTCGCGTTTTGTATTCTG TCGGCAT ctgaaaatcggatgaatttaatag | |
| | | HA2 Fwd: CGTGATTACAAGGATGACGA TGACAAG AGATCGCTCAACACAACCTTCAC | |
| | | HA2 Rev: tcacacaggaaacagctatgaccatgttat ACTCCACGGATGCTctgaat | |

*Continued*

| Allele | sgRNA sequence | Homology arm | | SEC used |
|--------|----------------|--------------|---|----------|
| *let-92* (*wow88[let-92::gfp^aid3xflag]*) | GAAAACGGCGATTTGAACGG**AGG** (pJM51) | HA1 Fwd: ttgtaaaacgacggccagtcgccggca CCTTCACGGAGGTCTTTCAC | | pJW1583 |
| | | HA1 Rev: CATCGATGCTCCTGAGGC TCCCGATGC TCCCAGGAAGTAGTCAGGCG TTCT | | |
| | | HA2 Fwd: CGTGATTACAAGGATGACGA TGAC AAGAGATAGatagatacctccgtt- caaatcg | | |
| | | HA2 Rev: ggaaacagctatgaccatgttatcg atttcgggaagtggtgaaaaggatg | | |
| *sas-4* (*wow32[zf::gfp3xflag::sas-4]*) | *GGA*AAACAACTTTGTTCCAG (pJF296) | HA1 Fwd: ttgtaaaacgacggccagtcgccgg caaattgtaaaatttggcgccttcaa | | pJF250 |
| | | HA1 Rev: CATCGATGCTCCTGAGGC TCCCGATGCTCCT TTTTTCCATTGAAACAATGTAGTC T | | |
| | | HA2 Fwd: CGTGATTACAAGGATGACGA TGACA AGAGATGAgaaattccaaccccttt | | |
| | | HA2 Rev: ggaaacagctatgaccatgttatcgat ttcaagatgctgctcctggatgt | | |

## Image acquisition

Embryos dissected from one-day old adults were mounted on a pad (3% agarose dissolved in M9) sandwiched between a microscope slide and no. 1.5 coverslip. Time-lapse images were acquired on a Nikon Ti-E inverted microscope (Nikon Instruments) equipped with a 1.5x magnifying lens, a Yoko-gawa X1 confocal spinning disk head, and an Andor Ixon Ultra back thinned EM-CCD camera (Andor), all controlled by NIS Elements software (Nikon). Images were obtained using a 60x Oil Plan Apochromat (NA = 1.4) or 100x Oil Plan Apochromat (NA = 1.45) objective. Z-stacks were acquired using a 0.5 µm step every minute. Images were adjusted for brightness and contrast using ImageJ software.

## Drug treatment

Drugs treatments were performed as previously described (*Yang and Feldman, 2015*). Briefly, embryos were mounted between a slide and coverslip, supported with 22.5 uM beads (Whitehouse Scientific), and bathed in an osmotically balanced control buffer (embryonic growth medium – EGM [*Shelton and Bowerman, 1996*]) supplemented with either 10% DMSO, 30 µM okadaic acid, or 60 µM rubratoxin A, or 200 µM flavopiridol. Embryos were laser permeabilized at appropriate times using a Micropoint dye laser (coumarin 435 nm) mounted on the spinning-disk confocal described above.

## Cell fusion experiments

Embryos were prepared and mounted the same way as described above for image acquisition. ABp and $P_2$ cells were fused using the Micropoint dye laser (coumarin 435 nm) and confocal described above.

### RNAi treatment

RNAi treatment was performed by feeding as previously described using *csnk-1*(RNAi), *gpr-1/2* (RNAi), and *let-92*(RNAi) expressing HT115 bacteria from the Ahringer RNAi library (*Ahringer, 2006*; *Fraser et al., 2000*; *Kamath et al., 2003*). L4 stage worms were grown on RNAi plates (NGM supplemented with IPTG and Ampicillin) at 25°C for 24 h-48h. RNAi plates were seeded with a bacterial culture grown overnight and subsequently grown 48 hr at room temperature protected from light.

### Image quantification

#### PCM volume measurements

PCM volume was measured from stacks of images taken through the ABp centrosome closest to the coverslip at each timepoint. Image stacks were first processed to eliminate the cytosolic background by subtracting the mean intensity of 10 random points in the cytoplasm at each plane and each timepoint. Image stacks were then thresholded using the Otsu method (ImageJ) to delimit the PCM structure. Volume measurements were performed using the 3D object counter imageJ plugin (*Bolte and Cordelières, 2006*). Only the volume of PCM that was connected to the centrioles was considered. Individualized packets that were physically separate from the centrioles were manually subtracted from the total PCM volume.

#### Intensity measurements

Total intensity was measured by defining an image stack 15 μm wide x 7.5 μm deep around the centrosome for each timepoint. Another stack of the exact same dimensions was generated in the cytoplasm. Both stacks were sum projected and the total intensity was measured by subtracting the total intensity of the cytoplasmic sum projection from the total intensity of the centrosome sum projection. Centrosomal intensity was calculated in the same way, but ROIs were selected manually following initial thresholding to specifically select for the centrosome and not the surrounding packets upon rupture. Only the PCM that was connected to the centrioles was considered in the intensity measurement. Packet intensity was determined by subtracting the intensity measurement for the centriole and contiguous PCM from the total intensity measurement. In *Figure 5G*, we accounted for the fact that *let-92* depletion results in centriole duplication defects in the one cell embryo (*Song et al., 2011*). In control embryos, we determined the average intensity of each of the two individual centriolar/centrosomal foci of either SPD-2 or SPD-5 at the end of disassembly (t = 5′). We compared this value to the average intensity of the single centrosomes in *let-92* depleted embryos at the end of disassembly (t = 15′). This type of measurement was in contrast to the total centriole/centrosome measurement shown in *Figure 5E and F*, which does not distinguish the two resulting centrioles/centrosomes in control conditions at the end of disassembly.

#### Timing of events

The different steps of disassembly were defined based on hallmarks of both volume and intensity measurements. 'Dissolution' was defined as the timepoint at which the first decrease in PCM intensity was detected, which corresponded to a decrease in SPD-2 intensity. 'Rupture' was defined as the timepoint at which holes first appear in the PCM, which corresponded to a drop in SPD-5 volume. Packet formation was defined as the timepoint at which individualized foci of SPD-5 appeared around the centrioles.

#### Statistics

Statistical analyses were performed using R and Prism (GraphPad software, La Jolla, Ca, USA). PCM size reported in *Figure 1* were statistically tested using an unpaired *t* test. All other data were analyzed using an ANOVA analysis followed by Tukey's multiple comparison test; P-values for the Tukey's multiple comparison test are reported in corresponding figure legends. To determine the percentage of SPD-5 overlapping with SPD-2, the area under the curve (AUC) of the mean intensity profile of SPD-5 was used. The portion of the SPD-5 overlapping with SPD-2 was defined as the SPD-5 AUC in the region defined by the half-max values of the SPD-2 mean intensity profile. The total portion of SPD-5 was defined as the AUC in the region defined by the half-max values of the SPD-5 mean intensity profiles. The percentage of overlap was then defined as the SPD-5 AUC in SPD-2 half-max interval divided by the SPD-5 AUC in the SPD-5 half-max interval.

## Acknowledgements

We thank Kevin O'Connell, Jyoti Iyer, Dan Dickinson, and Bob Goldstein for CRISPR advice and protocols. We also thank Tim Stearns, Ariana Sanchez, Maria Sallee, and members of the Feldman lab for helpful discussions about the manuscript. Some of the nematode strains used in this work were provided by the *Caenorhabditis* Genetic Center, which is funded by the NIH Office of Research Infrastructure Programs (P40 OD010440). This work was supported by a March of Dimes Basil O'Connor Starter Scholar Research Award and an NIH New Innovator Award DP2GM119136-01 awarded to JLF. JM is supported by an American Heart Postdoctoral Fellowship.

## Additional information

### Funding

| Funder | Grant reference number | Author |
| --- | --- | --- |
| March of Dimes Foundation | Basil O'Connor Starter Scholar Research Award | Jessica L Feldman |
| National Institutes of Health | DP2GM119136-01 | Jessica L Feldman |
| American Heart Association | Postdoctoral Fellowship | Jérémy Magescas |

The funders had no role in study design, data collection and interpretation, or the decision to submit the work for publication.

### Author contributions

Jérémy Magescas, Conceptualization, Formal analysis, Funding acquisition, Validation, Investigation, Visualization, Methodology, Writing—original draft, Writing—review and editing; Jenny C Zonka, Investigation, Methodology; Jessica L Feldman, Conceptualization, Resources, Supervision, Funding acquisition, Investigation, Visualization, Methodology, Writing—original draft, Project administration, Writing—review and editing

### Author ORCIDs

Jérémy Magescas ⓘD https://orcid.org/0000-0002-7832-0851
Jessica L Feldman ⓘD https://orcid.org/0000-0002-5210-5045

### Decision letter and Author response

Decision letter https://doi.org/10.7554/eLife.47867.023
Author response https://doi.org/10.7554/eLife.47867.024

## Additional files

### Supplementary files

• Transparent reporting form
DOI: https://doi.org/10.7554/eLife.47867.021

### Data availability

Data generated or analyzed during this study are included in the manuscript and supporting files.

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
