## [Decision Letter]

[Editors’ note: a previous version of this study was rejected after peer review, but the authors submitted for reconsideration. The first decision letter after peer review is shown below.]

Thank you for submitting your work entitled "A two-step mechanism for the inactivation of microtubule organizing center function at the centrosome" for consideration by *eLife*. Your article has been reviewed by three peer reviewers, and the evaluation has been overseen by a Reviewing Editor and a Senior Editor. The reviewers have opted to remain anonymous.

Our decision has been reached after consultation between the reviewers. Based on these discussions and the individual reviews below, we regret to inform you that your work will not be considered further for publication in *eLife*.

All three reviewers agreed that the present study is of potential interest. However, as you can see from the individual comments of these reviewers, they felt that the progress that has been achieved compared to the existing/published work was not sufficient. The reviewers thus agreed that the revision required for this manuscript to be acceptable for *eLife* (such as identification of PP2A substrates) is too substantial and beyond the scope of standard revision per *eLife*'s policy, as only revisions with straightforward experiments that can be carried out within approximately two months are encouraged by *eLife*.

We would be prepared to consider a substantially revised version as a new submission, if the mechanistic part of the paper were significantly extended, for example, by identifying new PP2A substrates or the regulatory details behind the two-step PCM disassembly.

Reviewer #1:

The manuscript by Magescas et al. reports on centrosome disassembly as cells exit mitosis. This process involves the reduction in PCM components and a corresponding reduction in centrosomal microtubules. The authors begin by defining the localization of 4 PCM components relative to the centriole to show a wider and wider distribution in the following order Spd-2, MZT-1, Spd-5 and then GIP-1. The authors then describe two different disassembly modes: Spd-2 and MZT-1 show a gradual disassembly, while Spd-5 and GIP-1 disassemble in "packets". The authors then use RNAi and chemical perturbations to show that centrosome disassembly requires microtubules to pull the packets away from the centrosome, and that PP2A is also required.

The topic is both interesting and important; centrosome disassembly, or dematuration, does not garner anywhere near the attention that centrosome maturation commands. Clearly dematuration is an active process and it is not simply PCM passively falling apart. This study, however, does not provide much new insight into disassembly. I outline this in detail below.

1) Centrosome fragmentation. I was surprise that there was no mention of the study from the Wadsworth lab from 2005 (Rusen and Wadsworth, 2005).. The Wadsworth lab showed the same phenomenon of disassembling the centrosome into packets, or fragments, and their ability to nucleate microtubules. While the previous study was done in another system, observation of packets in of themselves is not particularly novel. Another example that the authors did mention are the flares from *Drosophila* embryos. Here the authors properly reference the work by Megraw, but they improperly state that the flares are present throughout the cell cycle. Flares form specifically during mitotic exit and they also release packets into the cytoplasm. This was shown in Lerit et al. (2015).

2) The mechanism of centrosome disassembly. While the authors better describe the effects of reducing microtubule pulling forces (via Grp-1/2 loss), inhibition of PP2A function (drug treatment), and loss of Let-92 on centrosome disassembly, all of these experiments were previously reported in Enos et al., 2018. This previous study greatly reduces the enthusiasm for the current results. I think an advance to the field here would require the identification of PP2A targets.

3) Centrosome protein distribution. One exciting result was the finding that Spd-5 and Spd-2 do not occupy the same space in the centrosome. This could have important implications on how we view *C. elegans* PCM that might differ from current models. I was, however, surprised by the choice of imaging. The use of Structured Illumination Microscopy to define the regions occupied by centrosome protein is standard practice in the centrosome field. A localization study using SIM would be more appropriate, not simply because it is standard practice, but rather because SIM imaging has played a critical role in shaping current models and hypotheses regarding centrosome function.

Reviewer #2:

Magescas and coworkers investigate the regulation of PCM disassembly using dynamic imaging of the *C. elegans* embryo. The centrosome is the microtubule-nucleating organelle organized into distinct subconcentric domains. The stepwise assembly of the various centrosomal components resulting in the recruitment of γ-TURC complexes is well-studied based on work from a variety of systems.

In this study, the authors describe the macromolecular organization of the *C. elegans* centrosome, albeit at relatively low resolution compared to similar studies from cell lines and *Drosophila* embryos. Of note, the authors examine the less understood process of PCM disassembly and the associated role of protein phosphatase PP2A. Their work suggests disassembly of the retinue of PCM proteins by at least 2-steps. First, a slower-acting step, regulated in part by protein dephosphorylation, results in the partial dissolution of the PCM. Second, a faster-acting step involves active forces – presumably acting upon microtubules via protein complexes associated with the cortical membrane ("cortical forces").

This is an informative study that adds mechanistic understanding to the important process of centrosome inactivation via PP2A and cortical pulling forces. However, much of this work appears to largely support the conclusions of Enos et al., 2018 published in Biology Open. Nonetheless, the present work does delineate some differences with respect to the temporal requirement of phosphatases versus forces. Further, the authors describe impaired centriole separation in response to impairing the process of PCM disassembly, thereby adding some physiological significance. However, it is not apparent the present study presents enough of an advance to merit publication in *eLife*.

1) Would the authors please describe in more detail what they measure for their quantification of centrosome intensity? Is the ROI a defined shape/size? Do the authors include proteins that have ruptured and migrated away from the centriole? As written, the Materials and methods simply state the ROI "was selected manually". This merits a little more detail as a major conclusion is that changes in centrosomal intensity precede changes in volume (that is, dissolution occurs before rupture).

Reviewer #3:

Magescas et al. document the fine structure of the *C. elegans* centrosomal PCM, and two phases of its disassembly, in early embryonic cells. They show that the PCM consists of an inner core that includes the coiled coil proteins SPD-2 and SPD-5, along with γ-tubulin ring components, and an outer shell that includes SPD-5 and in part one γ-tubulin ring component (GIP-1) with a second γ-tubulin ring component (MTZ-1) extending beyond the inner core but extending only partially into the SPD-5/GIP-1 outer shell. The authors also document a two-step PCM dis-assembly process, with a PP2A phosphatase-dependent dissolution of some PCM components, and a cortical-force dependent rupture of the PCM that completes dis-assembly of the SPD-5 scaffold. These results significantly improve our understanding of *C. elegans* PCM structure and of PCM dis-assembly. However, before the manuscript can be considered suitable for publication in *eLife*, the authors need to address the following concerns.

1) In all of the figures, the authors show a dissolution of the PCM components SPD-2 and MTZ-1, without the rupture and packets observed for SPD-5 and GIP-1. However, the later time points do show two strong foci for SPD-2 and MTZ-1. Do these represent duplicated centrioles with associated PCM? The authors never described these foci in the text or figure legends and need to more fully describe the images they present for the reader to fully understand these results.

2) Similarly to the above comment (1), in the graphs showing quantification of packet intensity (Figure 2E), centrosomal intensity (Figure 3D) and volume at the centrosome (Figure 3E), are the authors excluding the bright foci for SPD-2 and MZT-1, or are the included? Because the graphs in some cases show signal levels close to zero, presumably these bright foci are somehow excluded, but the text and figure legends do not refer to this issue. Again, for clarity, the authors should more fully describe the image data in the figure legends and text. The Materials and methods seem to refer to this issue with respect to Figures 5E and 5F, but it is not clear to me exactly how these quantifications were done with respect to the two bright foci detected at later time points.

3) In Figure 2, the authors show that SPD-5 and GIP-1 are both found in packets during dis-assembly. However, Figure 2A shows SPD-5 present only in two foci in the "early packets", while there are several SPD-5 foci in early packets in Figure 2B. Thus the co-localization of SPD-5 and GIP-1 appears very limited early on. The authors do not refer to this in the text or legend. How extensive is the overlap between SPD-5 and GIP-1 in packets? Are there some packets with only one or the other, and some with both, and can the authors quantify this overlap? More information as to the exact nature of their co-localization would be helpful.

4) The authors refer to "rupture" and "packet formation" somewhat interchangeably. In the Figure 3 legend, they define these terms with reference to arrows, but they never refer to specific images that would clarify these terms. The authors should refer to specific images in the figures that show the "holes" in the SPD-5/GIP-1 matrix that constitute rupture, and images that show the appearance of individual packets. While the graphs in Figure 3 indicate rupture occurring at 6 minutes, SPD-5 is clearly fragmented at 5 minutes in Figure 3C, and something like "holes" are apparent at 4 minutes in Figure 3C. I am confused by the use of these two terms that seem to imply distinct phases in dis-assembly but also seem to be simply different stages of the same process. Some clarification of these terms would be helpful.

5) When addressing the cortical forces to promote packet formation/dis-assembly in Figure 4, the text refers to rupture starting at 6 minutes post-NEBD and packets forming at 8 minutes in the control/WT embryos (subsection “Cortical forces mediate the disassembly of the PCM and more specifically SPD-5”, second paragraph). However, the figure only goes to 7 minutes. Moreover, SPD-5 shows distinct puncta at 5 minutes in Figure 1A control, and what seem to be definite packets at 6 minutes. Thus the text description does not seem to be very consistent with what is shown in the figure. This needs clarification.

6) In the second paragraph of the subsection “Cortical forces mediate the disassembly of the PCM and more specifically SPD-5”, the authors refer to SPD-5 intensity and volume being increased or decreased by either grp-1/2 or *csnk-1* depletion. What is being increased and what is being decreased, and in what background. I could not understand this sentence as written, and it seemed to me that in Figure 4 the knockdowns led only to increases.

7) In the second paragraph of the Discussion, the authors state that removal of PCM proteins from the inner sphere weakens the remaining PCM, allow for rupture of the outer sphere. This is stated as if it is a clear conclusion, but what data supports this interpretation/conclusion, or is it a model and speculation?

8) The authors cite Enos et al. 2018 as showing the PP2A and cortical forces have been shown to be required for PCM dis-assembly. Thus the two factors that contribute to dis-assembly have already been published. However, Enos et al. examined only SPD-5 and no other PCM components, and did not distinguish between dissolution and dissolution as two distinct processes. They simply refer to PP2A and cortical forces as two independent contributions without assigning them to distinct aspects of dis-assembly. The work by Enos et al. 2018 thus could be viewed as making the results presented here as somewhat incremental and perhaps better suited for publication in a more specialized journal. But Magescas et al. do provide novel findings concerning the layered nature of the PCM in *C. elegans* and a substantially more in depth analysis of PCM dis-assembly. The authors need to more explicitly address how their analysis goes beyond what was shown by Enos et al., either in the Results or the Discussion. This could be done briefly but it seems to me such a comparison is warranted, given the substantial amount of overlap in the two manuscripts. Similarly, has any evidence for this partial independence of SPD-2 and SPD-5 localization been previously noted?

[Editors’ note: what now follows is the decision letter after the authors submitted for further consideration.]

Thank you for submitting your article "A two-step mechanism for the inactivation of microtubule organizing center function at the centrosome" for consideration by *eLife*. Your article has been reviewed by three peer reviewers, and the evaluation has been overseen by a Reviewing Editor and Anna Akhmanova as the Senior Editor. The reviewers have opted to remain anonymous.

The reviewers have discussed the reviews with one another and the Reviewing Editor has drafted this decision to help you prepare a revised submission.

As you can see in the individual comments, all reviewers agreed that the newly submitted version of the manuscript has greatly improved over the previous version. Their comments are mostly straightforward and thus can be addressed without the guide of consolidated review.

Thus, we would like to invite you to submit the revision. Please provide point-by-point responses to review comments, such that we can reach the decision in a timely manner, once we receive the revision.

Reviewer #1:

This revised manuscript from Magescas, Zonka and Feldman thoroughly addresses the comments from all three reviewers on the original submission. The authors have added a substantial amount of new data (including centrosomal localization of several more proteins--PLK-1, TAC-1, ZYG-9, LET-92, AIR-1 and TPXL-1); additional inhibitor studies that uncouple the regulation of SPD-2 and SPD-5; and an interesting cell fusion experiments that suggest assembly and disassembly states are stable and resistant to influence from cytoplasms that promote the opposite processes. The authors also now include reference to Megraw et al. from JCB in 2002, in which cell cycle-dependent centrosome fragmentation was observed, and provide a more detailed comparison of their work to a recent publication (Enos et al. in BIology Open in 2018). I agree with the authors that their analysis is substantially more extensive and goes well beyond what has been reported previously, significantly advancing our understanding of centrosome disassembly. In my opinion the revised manuscript warrants publication in *eLife*.

Reviewer #2:

The manuscript by Magescas et al. is a resubmission/new submission from the Feldman lab on the topic of centrosome disassembly during mitotic exit. The study uses live cell imaging and RNAi in *C. elegans* to describe a two-step disassembly model of PCM – dissolution followed by rupture. The study shows that some centrosome components such as Spd2 follow the dissolution pathway of disassembly, while others such as Spd5 undergo rupture to generate packets of smaller PCM assemblies that move outward away from the centrioles. Overall, I found the study quite a bit improved over the previous version with addition of new experiments, references and discussion. I am also now convinced that it does goes sufficiently beyond the Enos 2018 study. I have one major comment.

1) Related to the description of spheres and the use of confocal microscopy. An important point in this paper is to define regions of the centrosome occupied by different proteins. The authors describe three regions: the centriole, the inner sphere, and the outer sphere. The description of these regions, along with comparison with other systems in which SIM was used, leaves the reader with the impression that Spd2 (and other proteins) occupies a toroid (donut) area surrounding the centriole with edges of 0.51-1.15, while Spd5 occupies a toroid (donut) with edges of 1.15-1.66. I think that the imaging method and data presented do not support this view. The intensity profiles in Figure 1B suggests that roughly 80-90% of Spd5 is actually found within Spd2 localization area from -1.15 and +1.15. Thus one cannot conclude that the proteins are in distinct locations.

The response by the authors related to the reviewer's comment about why SIM was not used in this study does not acknowledge SIM imaging done in *Drosophila* embryos (also a live organism) to describe centrosome zones (Lerit et al. 2015) and many publication form the Raff lab (such as Conduit et al. 2014) and even SIM in the worm (PMID: 28103229). I can only conclude that SIM is very much possible and routine. Given that SIM was not performed in this study, the authors' description of these proteins as "distinct localization", “discrete layers”, and "novel protein territories" (rebuttal letter) are not supported by the data. I recommend the following:

- Change the description to 'outer edge of the small spd2 sphere' and 'the outer edge of the larger Spd5 sphere'. That way the author will not misunderstand the localization as toroids.

- Indicate the outer edge measurements on Figure 1D.

- Report the% of total Spd5 that overlaps with Spd2 (between -1.15 and 1.15) and that falls outside of the Spd2 sphere (<-1.15 + >1.15) using the area under the curve. This way the reader will get a better impression that there is actually only a small amount of Spd5 that does not overlap with Spd2.

- Soften the comparison between these measurement and previous SIM reports. The data in this study shows no toroid structure other than TPXL-1 and AIR-1; thus the direct comparison to Cnn, for example, is not supported. A comparison to other systems should be reserved for a future SIM study.

These changes will not impact any of the subsequent results. I think it is even more intriguing that two proteins that mostly occupy the same area (in my estimate from the linescans that 80% of Spd5 overlaps with Spd2) can behave so differently.

Reviewer #3:

Largely through live imaging and quantitative analysis, Magescas and coworkers present a detailed study of PCM disassembly and the unique behaviors of several PCM constituents. In this resubmission, the authors present a more comprehensive localization analysis of endogenously tagged PCM molecules at mitotic centrosomes and corresponding time-lapse imaging of their disassembly. Additional experimentation includes extended pharmacological inhibition to parse apart relative contributions of the pro-maturation kinase CDK1 versus phosphatase activity, which also complement RNAi studies. Elegant cell fusions between mitotic P2 cells and ABp cells undergoing PCM disassembly test the competency of centrosomes to respond to presumptive cytoplasmic cues.

Strengths of the revised work include beautiful, carefully documented imaging which supports the conclusions: (1) PCM disassembly occurs by two distinct steps. First, a slow dissolution step that requires PP2A/phosphatase activity. Next, a more rapid rupturing phase rips the PCM apart in a MT-derived force-dependent manner. (2) The mitotic *C. elegans* centrosome shows a subconcentric organization consisting of distinct and separable zones, similar to those reported in cultured human and *Drosophila* cells and *Drosophila* embryos. (3) Several pieces of evidence support the conclusion that disassembly of Spd2 and Spd5 may be uncoupled. This point is significant given their interdependent localization to the PCM. (4) Analysis of these and other PCM molecules support the overall conclusion that inner zone molecules and outer zone molecules tend to be differentially regulated during the process of PCM disassembly. (5) As previously noted, another strength of the work in the physiological link between timely PCM disassembly and proper centriole separation kinetics.

This work contains several interesting insights and opens up many lines of future investigation; however, in the absence of more mechanistic insight, it is not apparent the present study presents enough of an advance to merit publication in *eLife*.

---

## [Author Response]

[Editors’ note: the author responses to the first round of peer review follow.]

Reviewer #1:[…] The topic is both interesting and important; centrosome disassembly, or dematuration, does not garner anywhere near the attention that centrosome maturation commands. Clearly dematuration is an active process and it is not simply PCM passively falling apart. This study, however, does not provide much new insight into disassembly. I outline this in detail below.1) Centrosome fragmentation. I was surprise that there was no mention of the study from the Wadsworth lab from 2005 (Rusen and Wadsworth, 2005). The Wadsworth lab showed the same phenomenon of disassembling the centrosome into packets, or fragments, and their ability to nucleate microtubules. While the previous study was done in another system, observation of packets in of themselves is not particularly novel. Another example that the authors did mention are the flares from Drosophila embryos. Here the authors properly reference the work by Megraw, but they improperly state that the flares are present throughout the cell cycle. Flares form specifically during mitotic exit and they also release packets into the cytoplasm. This was shown in Lerit et al. 2015.

We thank the reviewer for this recommendation and now cite the Wadsworth paper in multiple places. We have also added a paragraph to our Discussion where we compare and contrast packets, flares, and centrosome fragments. These structures share some interesting similarities but also differ in some ways that we now highlight. In regards to the timing of flares, Megraw et al., 2002, stated that “The number of flares associated with centrosomes varies with the cell cycle. […] The intensity of flares is highest at cleavage telophase/interphase centrosomes and lowest at mitotic centrosomes, especially during metaphase and anaphase. In addition, Cnn appears to be associated with metaphase/anaphase centrosomes more tightly, giving the centrosome a more rounded appearance, with fewer of the projections that spawn flare particles seen on centrosomes at other phases of the cleavage cycle …”. The quantification shown in this paper indicates that flares are present during the entire span of the cell cycle, although the flares decrease in number and intensity in metaphase. In addition, the more recent study by Lerit et al., 2015, showed that flares are present and active throughout interphase. We have changed our statement concerning flares, highlighting the fact that packets are found during disassembly exclusively, while flares are mainly found during interphase and telophase.

2) The mechanism of centrosome disassembly. While the authors better describe the effects of reducing microtubule pulling forces (via Grp-1/2 loss), inhibition of PP2A function (drug treatment), and loss of Let-92 on centrosome disassembly, all of these experiments were previously reported in Enos et al., 2018. This previous study greatly reduces the enthusiasm for the current results. I think an advance to the field here would require the identification of PP2A targets.

We fully agree with the reviewer that we confirm the results that were published by Enos et al., 2017 while our experiments were underway. However, although some of the mechanistic aspects between the two papers appear similar on the surface, our manuscript has several important differences that we feel make it a more complete story of centrosome inactivation that will still appeal to a wide audience:

a) Enos et al. focused entirely on SPD-5, while our manuscript analyzes a more complete repertoire of PCM proteins and finds that SPD-5 removal is preceded by removal of several exclusively inner sphere proteins including SPD-2 and the γ-TuRC component MZT-1. Our data not only paint a more complete picture of centrosome inactivation, but also identify specific regulated steps in the disassembly process,

b) By looking at several PCM components, our study identifies novel protein territories within the *C. elegans* PCM. That known binding partners apparently localize to distinct regions of the PCM changes the way we think about the nature of PCM assembly and function,

c) We identify important behavioral differences in the manner by which PCM proteins are removed from the centrosome. Through this analysis we discovered a phase of gradual dissolution that precedes the formation of smaller sub-PCM ‘packets’ that retain MTOC potential. As Enos et al. only characterized SPD-5 and did not comment on its disassembly behavior, these modes of disassembly we find are completely novel and point to distinct regulatory mechanisms.

d) Although Enos et al. find a role for phosphatase activity and cortical forces in SPD-5 disassembly, again, we elucidate a more complete picture of how these mechanisms impact overall PCM disassembly. We find that centrosome inactivation is initiated by a dissolution behavior that appears to be controlled by phosphatase activity. This process potentially weakens the PCM, which is then fully cleared by cortical forces. While SPD-5 is dependent on cortical forces for its complete removal, we find that SPD-2 can be removed in the absence of cortical forces and instead relies on phosphatase activity for proper clearance.

e) We identify functional significance for the timely removal of PCM proteins; both subsequent centriole separation and accumulation of the next round of PCM are affected when SPD-5 is not properly cleared from the PCM,

f) Our study uses only endogenously tagged proteins for our analysis, rather than overexpressed transgenes, allowing us to more confidently describe endogenous behaviors and mechanisms.

These points have been added to the Discussion. We have also added additional experiments that further differentiate our paper from that of Enos. et al., 2017 and expand our understanding of PCM disassembly.

1) New localization studies reveal additional aspects of PCM organization and behavior: We imaged endogenously tagged PLK-1, TAC-1, ZYG-9, LET-92, AIR-1, and TPXL-1, measuring their distribution in the PCM (Figure 1 and Figure 1—figure supplement 1) as well as observing their disassembly behavior (Figure 1 and Figure 1—figure supplement 2). We found that these proteins also occupy distinct regions within the inner and outer sphere of PCM, notably with TAC-1 and binding partner ZYG-9 localization being distinguishable in the outer sphere, AIR-1 and TPXL-1 occupying an almost exclusively outer sphere donut localization, and AIR-1 and PLK-1 (two mitotic kinases) adopting complimentary localization patterns within the PCM. PLK-1 and TAC-1 were disassembled by dissolution, while ZYG-9, AIR-1, TPXL-1, and perhaps LET92 ruptured and formed packets. Intriguingly, AIR-1 and TPXL-1 appeared to rupture and spread onto microtubules several minutes before SPD-5 and GIP-1 rupture, suggesting that their reorganization might play key roles in the microtubule network reorganization leading to rupture of the remaining PCM proteins.

2) New inhibitor studies reveal the timing of phosphatase activity, a role for phosphatases in shaping PCM assembly, and uncouple the regulation of SPD-2 and SPD-5:

We wanted to test the model that phosphatase activity is present at the centrosome throughout mitosis and that disassembly is the result of continued phosphatase activity coupled to a decrease in kinase activity (Figure 6). We tested this model by observing the localization of LET-92, which we found to be associated with the centrosome throughout assembly and disassembly. We further tested this model by treating embryos at different times in mitosis with the CDK inhibitor flavopiridol (FP) and the broad-spectrum phosphatase inhibitor okadaic acid (OA). Our previous studies had indicated that FP treatment could force precocious disassembly of g-tubulin. We repeated this experiment in embryos expressing tagRFP::SPD-5; SPD-2::GFP and found that both of these proteins were also precociously disassembled by dissolution in the presence of FP. Notably, this precocious disassembly could be inhibited in the presence of both FP and OA, indicating that CDK activity on PCM assembly is normally opposed by phosphatase activity. Intriguingly, OA treatment alone stabilized the association of SPD-5 with the centrosome, but also forced the precocious removal of SPD-2 at NEBD. This experiment not only suggests that an OA sensitive phosphatase controls SPD-2 maintenance at the centrosome, but also further uncouples the regulation and centrosomal association of SPD-2 and SPD-5: SPD-2 leaves the centrosome without affecting the ability of SPD-5 to remain associated.

3) New cell fusion experiments indicate that the nature of the PCM changes throughout mitosis: PCM disassembly could merely be the exact converse of assembly, i.e. kinases catalyze the addition of PCM and phosphatases catalyze their subtraction. However, we found a role for cortical forces in outer sphere disassembly suggesting that phosphatases are not sufficient to remove total PCM from the centrosome, especially during the disassembly phase. These data as well as recent in vitro data support a model where the nature of the PCM changes throughout mitosis such that assembling and disassembling PCM are different. To test this model, we fused a cell with a disassembling centrosome (ABp) with a cell with an assembling centrosome (P_1_) to determine whether the cytoplasmic factors promoting either state had an effect on the other (Figure 6). We found PCM assembly and disassembly to be mutually resistant processes whereby the disassembling centrosome kept disassembling and the assembling centrosome kept assembling upon cell fusion. These results indicate that disassembling PCM is resistant to addition of new PCM by mitotic kinase and that assembling PCM is resistant to disassembly by disassembly competent cytoplasm. Intriguingly, once the old PCM was stripped from the disassembling centrosome, new PCM was precociously added in accordance with the influx of mitotic kinases upon cell fusion. These studies provide in vivo evidence for a change in the nature of the PCM throughout mitosis.

Together, these additional studies suggest that assembling PCM is regulated by both kinases and phosphatases that perhaps create a more mobile, fluid PCM state (Figure 7). With time, kinases are naturally inactivated in the cell cycle allowing phosphatase activity to dominate and remove a subset of inner sphere PCM. However, the nature of the PCM also changes such that the outer sphere PCM becomes resistant to removal by phosphoregulation, requiring mechanical disruption by cortical pulling forces for effective clearance.

3) Centrosome protein distribution. One exciting result was the finding that Spd-5 and Spd-2 do not occupy the same space in the centrosome. This could have important implications on how we view C. elegans PCM that might differ from current models. I was, however, surprised by the choice of imaging. The use of Structured Illumination Microscopy to define the regions occupied by centrosome protein is standard practice in the centrosome field. A localization study using SIM would be more appropriate, not simply because it is standard practice, but rather because SIM imaging has played a critical role in shaping current models and hypotheses regarding centrosome function.

We agree that SIM microscopy and other super-resolution techniques have been instrumental and a standard in the characterization of the organization of PCM proteins. To date, these studies have focused on characterizing PCM organization in cell lines, fixed samples, or isolated centrosomes (Fu et al., 2012, Lawo et al., 2012, Sonnen et al., 2012, Mennella et al., 2012). Here, we sought to understand PCM organization in a live organism by observing endogenously tagged proteins. Due to the nature of the sample, we were not able to use SIM microscopy or other super-resolution techniques. However, *C. elegans* PCM proteins displayed a dramatic difference in organization and behaviors such that we were still able to observe using conventional spinning-disk confocal microscopy. We agree that our conclusions are limited by our choice of imaging technique and have added a statement to this effect in the Discussion (second paragraph).

Reviewer #2:[…] This is an informative study that adds mechanistic understanding to the important process of centrosome inactivation via PP2A and cortical pulling forces. However, much of this work appears to largely support the conclusions of Enos et al., 2018 published in Biology Open. Nonetheless, the present work does delineate some differences with respect to the temporal requirement of phosphatases versus forces. Further, the authors describe impaired centriole separation in response to impairing the process of PCM disassembly, thereby adding some physiological significance. However, it is not apparent the present study presents enough of an advance to merit publication in eLife.1) Would the authors please describe in more detail what they measure for their quantification of centrosome intensity? Is the ROI a defined shape/size? Do the authors include proteins that have ruptured and migrated away from the centriole? As written, the Materials and methods simply state the ROI "was selected manually". This merits a little more detail as a major conclusion is that changes in centrosomal intensity precede changes in volume (that is, dissolution occurs before rupture).

We have now added additional details on the quantification of centrosomal intensity in the Materials and methods section (subsection “Intensity measurements”). We defined the ‘centrosomal’ region as the centrioles plus the region of the PCM that was physically connected to the centrioles as determined by our thresholding method. This region was manually selected for both the intensity and volume measurements. During the dissolution period, PCM intensity was effectively all of the imageable PCM as holes were not apparent in the PCM until several minutes later. Once rupture had occurred and holes became apparent, we only included the region of the PCM that was contiguous with the centrioles for intensity and volume measurements. The other non-contiguous regions were considered packets once they were clearly individualized from the centrioles and contiguous PCM.

Reviewer #3:[…] Before the manuscript can be considered suitable for publication in eLife, the authors need to address the following concerns.1) In all of the figures, the authors show a dissolution of the PCM components SPD-2 and MTZ-1, without the rupture and packets observed for SPD-5 and GIP-1. However, the later time points do show two strong foci for SPD-2 and MTZ-1. Do these represent duplicated centrioles with associated PCM? The authors never described these foci in the text or figure legends and need to more fully describe the images they present for the reader to fully understand these results.

The two bright foci are indeed the separating centrioles and newly forming centrosomes. We have indicated these structures in the figures with joined double arrows and have added a reference to them in the text (subsection “PCM proteins disassemble with different behaviors”, first paragraph) and figure legends.

2) Similarly to the above comment (1), in the graphs showing quantification of packet intensity (Figure 2E), centrosomal intensity (Figure 3D) and volume at the centrosome (Figure 3E), are the authors excluding the bright foci for SPD-2 and MZT-1, or are the included? Because the graphs in some cases show signal levels close to zero, presumably these bright foci are somehow excluded, but the text and figure legends do not refer to this issue. Again, for clarity, the authors should more fully describe the image data in the figure legends and text. The Materials and methods seem to refer to this issue with respect to Figures 5E and 5F, but it is not clear to me exactly how these quantifications were done with respect to the two bright foci detected at later time points.

Please see our response to Comment 1 of reviewer 2. For centrosomal intensity and volume, we manually measured the centrioles and physically contiguous PCM for our quantifications. Please note that our graphs show arbitrary units x 10^5^ for intensity, so the minimum values are on the order of 20,000. Our volume measurements decrease to <1-2 microns^3^ but are not zero. This measurement would be the expectation for naked centrioles given their dimensions and the diffraction limited approach we used for our quantifications. Compared to the very large size of the mature centrosomes, this small size is even more exaggerated.

3) In Figure 2, the authors show that SPD-5 and GIP-1 are both found in packets during dis-assembly. However, Figure 2A shows SPD-5 present only in two foci in the "early packets", while there are several SPD-5 foci in early packets in Figure 2B. Thus the co-localization of SPD-5 and GIP-1 appears very limited early on. The authors do not refer to this in the text or legend. How extensive is the overlap between SPD-5 and GIP-1 in packets? Are there some packets with only one or the other, and some with both, and can the authors quantify this overlap? More information as to the exact nature of their co-localization would be helpful.

We had accidentally switched SPD-5 and GIP-1 single panel images in Figure 2A, although they were correctly colored in the merge. The proteins do colocalize in early packets, there is just significantly more SPD-5 than GIP-1 in packets. This co-localization is also short lived as later packets lose GIP-1 as well as microtubule association (Figure 2A and 2B, late packet).

Quantification of the overlap proved difficult due to the relatively dim nature of packets compared to the surrounding cytoplasm, the ephemeral nature of packets, and the mixture of packets with persistent PCM, thus prompting us to look at microtubules and EBP-2/EB1 to further confirm whether packets retain MTOC potential.

4) The authors refer to "rupture" and "packet formation" somewhat interchangeably. In the Figure 3 legend, they define these terms with reference to arrows, but they never refer to specific images that would clarify these terms. The authors should refer to specific images in the figures that show the "holes" in the SPD-5/GIP-1 matrix that constitute rupture, and images that show the appearance of individual packets. While the graphs in Figure 3 indicate rupture occurring at 6 minutes, SPD-5 is clearly fragmented at 5 minutes in Figure 3C, and something like "holes" are apparent at 4 minutes in Figure 3C. I am confused by the use of these two terms that seem to imply distinct phases in dis-assembly but also seem to be simply different stages of the same process. Some clarification of these terms would be helpful.

Rupture is a gradual process that begins with the formation of holes in the PCM and ultimately results in packet formation. We see some variation in when rupture occurs and have included images in the figures that represent the mean timing for clarity. We have also included a more thorough description of what we define as rupture in the text (subsection “PCM proteins disassemble with different behaviors”, last paragraph). We have also eliminated packet formation as an independent step in the figures for clarity as we consider it as part of rupture.

5) When addressing the cortical forces to promote packet formation/dis-assembly in Figure 4, the text refers to rupture starting at 6 minutes post-NEBD and packets forming at 8 minutes in the control/WT embryos (subsection “Cortical forces mediate the disassembly of the PCM and more specifically SPD-5”, second paragraph). However, the figure only goes to 7 minutes. Moreover, SPD-5 shows distinct puncta at 5 minutes in Figure 1A control, and what seem to be definite packets at 6 minutes. Thus the text description does not seem to be very consistent with what is shown in the figure. This needs clarification.

See our response to point 4 above. We have clarified what we mean by rupture in the text and included images in the figures that reflect the mean value for rupture, although there is some variation in the timing of this process. We have eliminated ‘packet formation’ as a discrete step as we see it as part of the process of rupture.

6) In the second paragraph of the subsection “Cortical forces mediate the disassembly of the PCM and more specifically SPD-5”, the authors refer to SPD-5 intensity and volume being increased or decreased by either grp-1/2 or csnk-1 depletion. What is being increased and what is being decreased, and in what background. I could not understand this sentence as written, and it seemed to me that in Figure 4 the knockdowns led only to increases.

We have clarified this admittedly confusing statement to reflect the decrease in SPD-5 intensity following *csnk-1* depletion and the increase in SPD-5 intensity following *grp-1/2* depletion: “Interestingly, SPD-5 levels at the PCM were increased by *gpr-1/2* and decreased by *csnk-1* depletion (Figure 4A, Figure 4—figure supplement 1A).”

7) In the second paragraph of the Discussion, the authors state that removal of PCM proteins from the inner sphere weakens the remaining PCM, allow for rupture of the outer sphere. This is stated as if it is a clear conclusion, but what data supports this interpretation/conclusion, or is it a model and speculation?

This statement was indeed meant to be speculation. We have reworked our Discussion considerably and have removed this statement.

8) The authors cite Enos et al., 2018, as showing the PP2A and cortical forces have been shown to be required for PCM dis-assembly. Thus the two factors that contribute to dis-assembly have already been published. However, Enos et al. examined only SPD-5 and no other PCM components, and did not distinguish between dissolution and dissolution as two distinct processes. They simply refer to PP2A and cortical forces as two independent contributions without assigning them to distinct aspects of dis-assembly. The work by Enos et al., 2018 thus could be viewed as making the results presented here as somewhat incremental and perhaps better suited for publication in a more specialized journal. But Magescas et al. do provide novel findings concerning the layered nature of the PCM in C. elegans and a substantially more in depth analysis of PCM dis-assembly. The authors need to more explicitly address how their analysis goes beyond what was shown by Enos et al., either in the Results or the Discussion. This could be done briefly but it seems to me such a comparison is warranted, given the substantial amount of overlap in the two manuscripts. Similarly, has any evidence for this partial independence of SPD-2 and SPD-5 localization been previously noted?

Please see our response to Comment 2 of reviewer 1. We have also modified the Discussion to highlight the differences between our two papers. We are unaware of any other indication in the literature of the partial independence of SPD-2 and SPD-5 and see even more support for this independence in our newly added inhibitor experiments where SPD-2 can be forced from the centrosome with no impact on SPD-5 localization.

[Editors' note: the author responses to the re-review follow.]

Reviewer #2:[…] 1) Related to the description of spheres and the use of confocal microscopy. An important point in this paper is to define regions of the centrosome occupied by different proteins. The authors describe three regions: the centriole, the inner sphere, and the outer sphere. The description of these regions, along with comparison with other systems in which SIM was used, leaves the reader with the impression that Spd2 (and other proteins) occupies a toroid (donut) area surrounding the centriole with edges of 0.51-1.15, while Spd5 occupies a toroid (donut) with edges of 1.15-1.66. I think that the imaging method and data presented do not support this view. The intensity profiles in Figure 1B suggests that roughly 80-90% of Spd5 is actually found within Spd2 localization area from -1.15 and +1.15. Thus one cannot conclude that the proteins are in distinct locations.

We did not intend to give the impression that SPD-2 and SPD-5 form interlocking toroids, but rather that both SPD-2 and SPD-5 localize to the inner sphere and that only SPD-5 extends into the outer sphere. With our imaging approach, we cannot resolve whether there is a hole in either localization pattern where the centrioles reside. The only toroids we observed was with AIR-1 and TPXL-1, as has been previously reported. As measured by the distance from center at half maximum intensity, SPD-2 falls in a region between -0.575 – 0.575 ± 0.02 and SPD-5 falls in a region between -0.83 – 0.83 ± 0.03 µm. 77.8 ± 0.8% of SPD-5 is found in the SPD-2 delimited region. These values are now reported in the second paragraph of the subsection “*C. elegans* PCM is organized into an inner and outer sphere”. We have also changed our graph in Figure 1C and our cartoon in Figure 1D to make these points more clear.

The response by the authors related to the reviewer's comment about why SIM was not used in this study does not acknowledge SIM imaging done in Drosophila embryos (also a live organism) to describe centrosome zones (Lerit et al. 2015) and many publication form the Raff lab (such as Conduit et al. 2014) and even SIM in the worm (PMID: 28103229). I can only conclude that SIM is very much possible and routine. Given that SIM was not performed in this study, the authors' description of these proteins as "distinct localization", “discrete layers”, and "novel protein territories" (rebuttal letter) are not supported by the data. I recommend the following:- Change the description to 'outer edge of the small spd2 sphere' and 'the outer edge of the larger Spd5 sphere'. That way the author will not misunderstand the localization as toroids.

We have changed our description to read:

“SPD-2 and SPD-5 localization at the PCM is co-dependent (Hamill et al., 2002; Kemp et al., 2004; Pelletier et al., 2004), however these proteins displayed distinct outer localization boundaries within the PCM; both SPD-2 and SPD-5 localized to a more proximal region surrounding the centrioles (distance from center at half maximum intensity for SPD-2: -0.575 – 0.575 ± 0.02 µm; 77.8 ± 0.8% of total SPD-5 overlapping with SPD-2 in this region), and SPD-5 extended more distally to a region lacking SPD-2 (distance from center at half maximum intensity for SPD-5: -0.83 – 0.83 ± 0.03 µm; Figure 1B-C). Based on the outer edge of these two matrix proteins, we divide the PCM into an ‘inner’ and ‘outer’ sphere, with the smaller inner sphere defined by the outer edge of SPD-2 localization and the larger outer sphere defined by the outer edge of SPD-5

localization (Figure 1D)”.

- Indicate the outer edge measurements on Figure 1D.

We have now reported this measurement (see above, and Figure 1 legend). We have also changed our graph in Figure 1C to more clearly depict the outer edges of localization (i.e. distance from center at half max intensity).

- Report the% of total Spd5 that overlaps with Spd2 (between -1.15 and 1.15) and that falls outside of the Spd2 sphere (<-1.15 + >1.15) using the area under the curve. This way the reader will get a better impression that there is actually only a small amount of Spd5 that does not overlap with Spd2.

We now report the percent total in the second paragraph of the subsection “*C. elegans* PCM is organized into an inner and outer sphere” (77.8 ± 0.8% of total SPD-5 overlapping with SPD-2 in this region).

- Soften the comparison between these measurement and previous SIM reports. The data in this study shows no toroid structure other than TPXL-1 and AIR-1; thus the direct comparison to Cnn, for example, is not supported. A comparison to other systems should be reserved for a future SIM study.

We have softened our comparison:

“Although our imaging approach did not allow us to resolve toroidal localization patterns of the majority of the PCM proteins we analyzed, the boundaries of PCM protein localization follows the general pattern of the predicted orthologs in *Drosophila* and human cells”.